

**Simulating CH$_4$ and CO$_2$ over South and East Asia using the zoomed**
**chemistry transport model LMDzINCA**
Xin Lin[1], Philippe Ciais[1], Philippe Bousquet[1], Michel Ramonet[1], Yi Yin[1], Yves Balkanski[1],
Anne Cozic[1], Marc Delmotte[1], Nikolaos Evangeliou[2], Nuggehalli K. Indira[3], Robin
Locatelli[1a], Shushi Peng[4], Shilong Piao[4], Marielle Saunois[1], Panangady S. Swathi[3], Rong
Wang[1], Camille Yver-Kwok[1], Yogesh K. Tiwari[5], Lingxi Zhou[6]
Affiliations:
[1]Laboratoire des Sciences du Climat et de l'Environnement, LSCE-IPSL (CEA-CNRS-
UVSQ), Université Paris-Saclay, 91191 Gif-sur-Yvette, France
[2]Norwegian Institute for Air Research (NILU), Department of Atmospheric and Climate
Research (ATMOS), Kjeller, Norway
[3]CSIR Fourth Paradigm Institute (formerly CSIR Centre for Mathematical Modelling and
Computer Simulation), NAL Belur Campus, Bengaluru 560 037, India
[4]Sino-French Institute for Earth System Science, College of Urban and Environmental
Sciences, Peking University, Beijing 100871, China
[5]Centre for Climate Change Research, Indian Institute of Tropical Meteorology, Pune, India
[6]Chinese Academy of Meteorological Sciences (CAMS), China Meteorological
Administration (CMA), Beijing, China
*Correspondence to:* X. Lin (xin.lin@lsce.ipsl.fr)
[a] Now at: AXA Global P&C, Paris, France



**Abstract**
The increasing availability of atmospheric measurements of greenhouse gases (GHGs) from
surface stations can improve the retrieval of their fluxes at higher spatial and temporal
resolutions by inversions, provided that chemistry transport models are able to properly
represent the variability of concentrations observed at different stations. South and East Asia
(SEA) is a region with large and very uncertain emissions of carbon dioxide ($CO_2$) and
methane ($CH_4$), the most potent anthropogenic GHGs. Monitoring networks have expanded
greatly during the past decade in this region, which should contribute to reducing
uncertainties in estimates of regional GHG budgets. In this study, we simulate concentrations
of $CH_4$ and $CO_2$ using a zoomed version of the global chemistry transport model LMDzINCA
during the period 2006–2013. The zoomed version has a fine horizontal resolution of ~0.66°
in longitude and ~0.51° in latitude over SEA and a coarser resolution elsewhere. The
concentrations of $CH_4$ and $CO_2$ simulated from the zoomed model (abbreviated as 'ZASIA')
are compared to those from the same model but with a uniform regular grid of 2.50° in
longitude and 1.27° in latitude (abbreviated as 'REG'), both having the same vertical 19
sigma pressure levels and prescribed with the same biogenic and anthropogenic fluxes.
Model performance is evaluated for annual gradients between sites, seasonal, synoptic and
diurnal variations, against a new dataset including 30 surface stations over SEA and adjacent
regions. Our results show that, when prescribed with identical surface fluxes, compared to
REG, the ZASIA version moderately improves the representation of $CH_4$ mean annual
gradients between stations as well as the seasonal and synoptic variations of this trace gas
within the zoomed region. This moderate improvement probably results from reduction of
representation errors and a better description of the $CH_4$ concentration gradients related to the
skewed spatial distribution of surface $CH_4$ emissions, suggesting that the zoom transport
model will be better suited for inversions of $CH_4$ fluxes in SEA. With the relatively coarse
vertical resolution and low-frequency (monthly) prescribed fluxes, the model generally does
not capture the diurnal cycle of $CH_4$ at most stations even with its zoomed configuration,
emphasizing the need to increase the vertical resolution, and to improve parameterizations of
turbulent diffusion in the planetary boundary layer and deep convection during the monsoon
period. The model performance for $CH_4$ is better than that for $CO_2$ at any temporal scale,
likely due to inaccuracies in the $CO_2$ fluxes prescribed in this study.



## 1 Introduction

Despite attrition in the global network of greenhouse gas (GHG) monitoring stations (Houweling et al., 2012), new surface stations have been installed since the late 2000s in the northern industrialized continents such as Europe (e.g., Aalto et al., 2007; Biraud et al., 2000; Haszpra, 1995; Levin et al., 1995; Lopez et al., 2015; Popa et al., 2010), North America (e.g., Bakwin et al., 1998; Dlugokencky et al., 1995; Miles et al., 2012), and Northeast Asia (e.g., Fang et al., 2014; Sasakawa et al., 2010; Wada et al., 2011; Winderlich et al., 2010). In particular, the number of continuous monitoring stations over land has increased (e.g., Aalto et al., 2007; Bakwin et al., 1998; Lopez et al., 2015; Winderlich et al., 2010) given that more stable and precise instruments are available (e.g., Yver Kwok et al., 2015). These observations can be assimilated in inversion frameworks that combine them with a chemistry transport model and prior knowledge of fluxes to optimize GHG sources and sinks (Berchet et al., 2015; Bergamaschi et al., 2010, 2015, Bousquet et al., 2000, 2006; Bruhwiler et al., 2014; Gurney et al., 2002; Peters et al., 2010; Rödenbeck et al., 2003). Given the increasing observation availability, GHG budgets are expected to be retrieved at finer spatial and temporal resolutions by atmospheric inversions if the atmospheric GHG variability can be properly modeled at theses scales. A first step of any source optimization is to evaluate the ability of chemistry transport models to represent the variabilities of GHG concentrations, as transport errors are recognized as one of the main uncertainties in atmospheric inversions (Locatelli et al., 2013).

Many studies have investigated regional and local variations of atmospheric GHG concentrations using atmospheric chemistry transport models, with spatial resolutions ranging 100–300 km for global models (e.g., Chen and Prinn, 2005; Feng et al., 2011; Law et al., 1996; Patra et al., 2009a, 2009b) and 10–100 km for regional models (e.g., Aalto et al., 2006; Chevillard et al., 2002; Geels et al., 2004; Wang et al., 2007). Model intercomparison experiments showed that the atmospheric transport models with higher horizontal resolutions are more capable of capturing the observed short-term variability at continental sites (Geels et al., 2007; Law et al., 2008; Maksyutov et al., 2008; Patra et al., 2008; Saeki et al., 2013), due to reduction of representation errors (point measured versus gridbox-averaged modeled concentrations), improved model transport, and more detailed description of surface fluxes and topography (Patra et al., 2008). However, a higher horizontal model resolution also





demands high-quality meteorological forcings and prescribed surface fluxes as boundary
conditions (Locatelli et al., 2015).
Two main approaches have been deployed, in an Eulerian modeling context, to address the
need for high-resolution transport modeling of long-lived GHGs. The first approach is to
define a high-resolution grid mesh in a limited spatial domain of interest, and to nest it within
a global model with varying degrees of sophistication to get boundary conditions for the
GHGs advected inside/outside the regional domain (Bergamaschi et al., 2005, 2010; Krol et
al., 2005; Peters et al., 2004). The second approach is to stretch the grid of a global model
over a specific region (the so-called 'zooming') while maintaining all parameterizations
consistent (Hourdin et al., 2006). For the former approach, several nested high-resolution
zooms can be embedded into the same model (Krol et al., 2005) to focus on different regions.
The 'zooming' approach has the advantage to avoid the nesting problems (e.g., tracers
discontinuity, transport parameterization inconsistency) at the boundaries between a global
and a regional model. In this study, we use the zooming capability of the LMDz model
(Hourdin et al., 2006).
South and East Asia (hereafter 'SEA') has been the largest anthropogenic GHG emitting
region since the mid 2000s due to its rapid socioeconomic development (Boden et al., 2015;
Olivier et al., 2015; Le Quéré et al., 2015; Tian et al., 2016). Compared to Europe and North
America where sources and sinks of GHGs are partly constrained by atmospheric
observational networks, the quantification of regional GHG fluxes over SEA from
atmospheric inversions remains uncertain because of the low density of surface observations
(e.g., Patra et al., 2013; Swathi et al., 2013; Thompson et al., 2014, 2016). During the past
decade, a number of new surface stations have been deployed (e.g., Fang et al., 2016, 2014;
Ganesan et al., 2013; Lin et al., 2015; Tiwari and Kumar, 2012), which have the potential to
provide new and useful constraints on estimates of GHG fluxes in this region. However,
modeling GHG concentrations at these stations is challenging since they are often located in
complex terrains (e.g. coasts or mountains) or close to large local sources of multiple origins.
To fully take advantage of the new surface observations in SEA, forward modeling studies
based on high-resolution transport models are needed to evaluate the ability of the inversion
framework to assimilate such new observations.





In this study, we apply the chemistry transport model LMDzINCA (Folberth et al., 2006;
Hauglustaine et al., 2004; Hourdin et al., 2006; Szopa et al., 2013) zoomed down to a
horizontal resolution of ~50km over SEA to simulate the variations of $CH_4$ and $CO_2$ during
the period 2006–2013. The model performance is evaluated against observations from 20
flask and 13 continuous stations within and around the zoomed region. The variability of the
observed or simulated concentrations at each station is decomposed for evaluation at different
temporal scales, namely: the annual mean gradients between stations, the seasonal cycle, the
synoptic variability and the diurnal cycle. For comparison, a non-zoomed version of the same
transport model is also run with the same set of surface fluxes and the same vertical pressure
levels, in order to estimate the improvement brought by the zoomed configuration. The
detailed description of the observations and the chemistry transport model is presented in
Section 2, together with the prescribed $CH_4$ and $CO_2$ surface fluxes that force the simulations,
as well as the metrics used to quantify the model performance. An evaluation of the
simulations performed is presented and discussed in Section 3, showing capabilities of the
transport model to representthe  annual gradients between stations, and the seasonal, synoptic,
and diurnal variations. Conclusions and implications drawn from this study are given in
Section 4.
**2 Data and Methods**
**2.1 Model description**
2.1.1 LMDzINCA
The LMDzINCA model couples a general circulation model developed at the Laboratoire de
Météorologie Dynamique (LMD; Hourdin et al., 2006), and a global chemistry and aerosol
model INteractions between Chemistry and Aerosols (INCA; Folberth et al., 2006;
Hauglustaine et al., 2004). A more recent description of LMDzINCA is presented in Szopa et
al. (2013). To simulate $CH_4$ and $CO_2$ concentrations, we run a regular version of the model
with a horizontal resolution of 2.5° (i.e., 144 model grids) in longitude and 1.27° (i.e., 142
model grids) in latitude (hereafter this version is abbreviated as 'REG') and a zoomed version
with the same number of grid boxes, but a resolution of ~0.66° in longitude and  ~0.51° in
latitude in a region of 50–130°E and 0–55°N centered over India and China (hereafter this
version is abbreviated as 'ZASIA') (Figure 1; see also Wang et al., 2014, 2016). It means that,

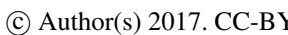



in terms of the surface area, a gridcell from REG roughly contains 9 grid-cells from ZASIA
within the zoomed region. Both model versions have 19 sigma-pressure layers extending
from the surface up to about 3.8 hPa, corresponding to a vertical resolution of about 300–500
m in the planetary boundary layer (first level at 70 m height) and about 2 km at the
tropopause (with 7–9 levels located in the stratosphere) (Hauglustaine et al., 2004). Vertical
diffusion and deep convection are parameterized following the schemes of Louis (1979) and
Tiedtke (1989), respectively. The simulated horizontal wind vectors ($u$ and $v$) are nudged
towards the 6-hourly European Center for Medium Range Weather Forecast (ECMWF)
reanalysis dataset (ERA-I) in order to simulate the observed large scale advection (Hourdin
and Issartel, 2000).
The atmospheric concentrations of hydroxyl radicals (OH), the main sink of atmospheric $CH_4$,
are produced from a simulation at a horizontal resolution of 3.75° in longitude (i.e., 96 model
grids) and 1.9° in latitude (i.e., 95 model grids) with the full INCA tropospheric
photochemistry scheme (Folberth et al., 2006; Hauglustaine et al., 2004, 2014). The OH
fields are climatological monthly data, and are interpolated onto the regular and zoomed
model grids, respectively. The magnitude of the OH fields are scaled globally in order that
the interannual variation of the simulated $CH_4$ growth rate agrees well with the observed
value (GLOBALVIEW-CH4, 2009). It should be noted that the spatiotemporal distributions
of the OH concentrations have large uncertainties and vary greatly among different chemical
transport models, therefore the choice of the OH fields may affect the evaluation for $CH_4$
(especially in terms of the annual gradients between stations and the seasonal cycles). In this
study, as we focus more on the improvement of model performance gained from refinement
of the horizontal resolution rather than model-observation misfits, the influences of OH
variations are assumed to be very small given that the OH fields for both ZASIA and REG
are interpolated from a lower model resolution and thus don't show much difference between
the two model versions.
The $CH_4$ and $CO_2$ concentrations are simulated over the period 2000–2013 with both REG
and ZASIA. The first six years (2000–2005) of the simulations are considered as model spin-
up, thus we only compared the simulated $CH_4$ and $CO_2$ concentrations with observations
during 2006–2013. The spin-up time of 6 years may appear short for $CH_4$ given the ~9-year
lifetime of $CH_4$ in the atmosphere, but simulations are started in 2000 from an initial state





with already realistic and almost balanced atmosphere between sources and sinks. The time
step of model outputs is hourly.
2.1.2 Prescribed $CH_4$ and $CO_2$ surface fluxes
The prescribed $CH_4$ and $CO_2$ surface fluxes used as model inputs are presented in Table 1.
We simulate the $CH_4$ concentration fields using a combination of the following datasets: (1)
the interannually varing anthropogenic emissions obtained from the Emission Database for
Global Atmospheric Research (EDGAR) v4.2 FT2010 product (http://edgar.jrc.ec.europa.eu),
including emissions from rice cultivation with the seasonal variations based on Matthews et
al. (1991) imposed to the original yearly data; (2) climatogical wetland emissions based on
the scheme developed by Kaplan et al. (2006); (3) interannually and seasonally varying
biomass burning emissions from Global Fire Emissions Database (GFED) v3.0 product (van
der Werf et al., 2010; http://www.globalfiredata.org/data.html), (4) climatological termite
emissions (Sanderson, 1996), (5) climatological ocean emissions (Lambert and Schmidt,
1993), and (6) climatological soil uptake (Ridgwell et al., 1999). Based on these emission
fields, the global $CH_4$ emissions in 2010 are 550 TgCH4/yr, and 194 TgCH4/yr over the
zoomed region. For the years over which $CH_4$ anthropogenic emissions (namely, the years
2011–2013) and biomass burning emissions (the years 2012–2013) were not available from
the data sources when the simulations were performed, we use emissions for the years 2010
and 2011 respectively.
The prescribed $CO_2$ fluxes used to simulate the concentration fields are based on the
following datasets: (1) three variants (hourly, daily, and monthly means) of interannually
varying fossil fuel emissions produced by the Institut für Energiewirtschaft und Rationelle
Energieanwendung (IER), Universität Stuttgart on the basis of EDGARv4.2 product
(hereafter IER-EDGAR, http://carbones.ier.uni-stuttgart.de/wms/index.html) (Pregger et al.,
2007); (2) interannually and seasonally varying biomass burning emission from GFEDv3.1
(van der Werf et al., 2010; http://www.globalfiredata.org/data.html); (3) interannually and
hourly varying terrestrial biospheric fluxes produced from outputs of the Organizing Carbon
and Hyrology in Dynamic EcosystEm (ORCHIDEE) model; and (4) interannually and
seasonally varying air-sea $CO_2$ gas exchange maps developed by NOAA's Pacific Marine
Environmental Laboratory (PMEL) and Atlantic Oceanographic and Meteorological
Laboratory (AOML) groups (Park et al., 2010). Here ORCHIDEE runs with the trunk version



r1882            (source        code       available            at
https://forge.ipsl.jussieu.fr/orchidee/browser/trunk#ORCHIDEE with the revision number of
r1882), using the same simulation protocol as the SG3 simulation in MsTMIP project
(Huntzinger et al., 2013). The climate forcing data are obtained from CRUNCEP v5.3.2,
while the yearly land use maps, soil map and other forcing data (e.g., monthly $CO_2$
concentrations) are as described in Wei et al. (2014). The sum of global net $CO_2$ surface
fluxes in 2010 are 6.9 PgC/yr, and 3.9 PgC/yr over the zoomed region. Like for $CH_4$, we use
$CO_2$ biomass burning emissions in the year 2011 to represent emissions in the years 2012 and
2013. For the $CO_2$ fossil fuel emissions, the IER-EDGAR product is only available until 2009.
To generate the emission maps for the years 2010–2013, we scaled the emission spatial
distribution in 2009 using the global totals for these years based on the EDGAR v4.2 FT2010
datasets. The detailed information for each surface flux is listed in Table 1.
**2.2 Atmospheric $CH_4$ and $CO_2$ observations**
The simulated $CH_4$ and $CO_2$ concentrations are evaluated against observations from 20 flask
and 13 continuous surface stations within and around the zoomed region (Figure 1), operated
by different programs and organizations (Table 2). The stations where flask observations are
published (12 stations) mainly belong to the cooperative program organized by the NOAA
Earth    System    Research    Laboratory    (NOAA/ESRL,    available    at
ftp://aftp.cmdl.noaa.gov/data/trace_gases/). We also use flask obervations from stations
operated by China Meterological Administration (CMA, China) (the JIN, LIN and LON
stations, see also Fang et al., 2014), Commonwealth Scientific and Research Organization
(CSIRO, Australia) (the CRI station, Bhattacharya et al., 2009, available at
http://ds.data.jma.go.jp/gmd/wdcgg/), Indian Institute of Tropical Meteorology (IITM, India)
(the SNG station, see also Tiwari et al., 2014), and stations from the Indo-French cooperative
research program (the HLE, PON and PBL stations, Lin et al., 2015; Swathi et al., 2013). All
the $CH_4$ ($CO_2$) flask measurements are reported on or linked to the NOAA2004
(WMOX2007) calibration scale, which guarantees comparability between stations in terms of
annual means.
The continuous $CH_4$ and $CO_2$ measurements are obtained from 13 stations operated by Korea
Meteorological Administration (KMA, Korea) (the AMY, GSN and KIS stations), Aichi Air
Environment Division (AAED, Japan) (the MKW station), Japan Meteorological Agency



(JMA) (the MNM, RYO and YON stations), National Institute for Environmental Studies
(NIES, Japan) (the COI and HAT stations), Agency for Meteorology, Climatology and
Geophysics (BMKG, Indonesia) and Swiss Federal Laboratoires for Materials Testing and
Research (Empa, Switzerland) (the BKT station). These datasets are available from the World
Data Center for Greenhouse Gases (WDCGG, http://ds.data.jma.go.jp/gmd/wdcgg/). Besides,
continuous $CH_4$ and $CO_2$ measurements are also available from HLE and PON that have been
maintained by the Indo-French cooperative research program between LSCE in France and
IIA and CSIR4PI in India (Table 2). All the continuous $CH_4$ ($CO_2$) measurements used in this
study are reported on or traceable to the NOAA2004 (WMOX2007) scale except AMY, COI
and HAT. The $CO_2$ continuous measurements at COI are reported on the NIES95 scale,
which is 0.10 to 0.14 ppm lower than WMO in a range between 355 and 385 ppm (Machida
et al., 2009). The $CH_4$ continuous measurements at COI and HAT are reported on the NIES
scale, with a conversion factor to WMO scale of 0.9973 (JMA and WMO, 2014). For AMY,
the $CH_4$ measurements over most of the study period are reported on the KRISS scale but
they are not traceable to the WMO scale (JMA and WMO, 2014); therefore, we discarded
this station from the subsequent analyses of the $CH_4$ annual gradients between stations. Note
that most of the stations where continuous observations are available are located on the east
part of the zoomed region, with the exception of HLE, PON and BKT. The stations used in
this study span a large range of geographic locations (marine, coastal, mountain or
continental) with polluted and non-polluted environments. Both flask and continuous
measurements are used to evaluate the model's ability in representing the annual gradient
between stations, the seasonal cycle and the synoptic variability for $CH_4$ and $CO_2$. The
continuous measurements are also used to analyze the diurnal cycle for these two gases.
To evaluate the model performance with regards to vertical transport, we also use
observations of the $CO_2$ vertical profiles from passenger aircraft from the Comprehensive
Observation Network for TRace gases by AIrLiner (CONTRAIL) project (Machida et al.,
2008, http://www.cger.nies.go.jp/contrail/index.html). This dataset provides high-frequency
$CO_2$ measurements made by on-board continuous $CO_2$ measuring equipments (CMEs) during
commercial airflights between Japan and other Asian countries. The CONTRAIL data are
reported on the NIES95 scale, which is 0.10 to 0.14 ppm lower than WMO in a range
between 355 and 385 ppm (Machida et al., 2009). In this study, we select from the
CONTRAIL dataset all the $CO_2$ vertical profiles over SEA during the ascending and



descending flights for the period 2006–2011, which provided 1808 vertical profiles over a
total of 32 airports (Figure S1 and S2).
**2.3 Sampling methods and data processing**
The model outputs are sampled at the nearest gridpoint and vertical level to each station for
both REG and ZASIA. For flask stations, the model outputs are extracted at the exact hour
when each flask sample was taken. For continuous stations below 1000 m.a.s.l., since both
REG and ZASIA cannot reproduce accurately the nighttime $CH_4$ and $CO_2$ accumulation near
the ground as in most transport models (Geels et al., 2007), only afternoon (12:00–15:00 LST)
data are retained for further analyses of the annual gradients, the seasonal cycle and the
synoptic variability. For continuous stations above 1000 m.a.s.l. (only HLE in this study),
nighttime (00:00–3:00 LST) data are retained, to avoid sampling local air masses advected by
upslope winds from nearby valleys. During daytime, the local valley ascendances and the
complex terrain mesoscale circulations cannot be captured by a global transport model.
The curve-fitting routine (CCGvu) developed by NOAA Climate Monitoring and Diagnostic
Laboratory (NOAA/CMDL) is applied to the modelled and observed $CH_4$ and $CO_2$ time
series to extract the annual means, monthly smoothed seasonal cycles and synoptic variations
(Thoning et al., 1989). For each station, a smoothed function is fitted to the observed or
modelled time series, which consists of a first-order polynomial for the growth rate, two
harmonics for the annual cycle (Levin et al., 2002; Ramonet et al., 2002), and a low-pass
filter with 80 and 667 days as short-term and long-term cutoff values, respectively (Bakwin et
al., 1998). The annual means and the mean seasonal cycle are calculated from the smoothed
curve and harmonics, while the synoptic variations are defined as the residuals between the
original data and the smoothed fitting curve. Note that we have excluded the observations
lying beyond three standard deviations of the residuals around the fitting curve, which are
likely to be outliers that are influenced by local fluxes. More detailed descriptions about the
curve-fitting procedures and the set-up of parameters can be found in Section 2.3 of Lin et al.

299    (2015).

For the $CO_2$ vertical profiles from the CONTRAIL passenger aircraft programme, since $CO_2$
data have been continuously taken every 10 seconds by the onboard CMEs, we average the
observed and corresponding simulated $CO_2$ time series into altitude bins of 1km from the
surface to the upper troposphere. We also divide the whole study area into four major





subregions for which we group all available CONTRAIL $CO_2$ profiles (Figure S1), namely
East Asia (EAS), the Indian sub-continent (IND), Northern Southeast Asia (NSA) and
Southern Southeast Asia (SSA). Given that there are model-observation discrepancies in $CO_2$
growth rates as well as misfits of absolute $CO_2$ concentrations, the observed and simulated
CONTRAIL time series have been detrended before comparisons of the vertical gradients. To
this end, over each subregion, we detrend for each altitude bin the observed and simulated
$CO_2$ time series, by applying the respective linear trend fit to the observed and simulated $CO_2$
time series of the altitude bin 3–4 km. This altitude bin is thus chosen as reference due to
greater data availability compared to other altitudes, and because this level is outside the
boundary layer where aircraft $CO_2$ data are more variable and influenced by local sources
(e.g. airports and nearby cities). The detrended $CO_2$ (denoted as $\Delta CO_2$) referenced to the 3-4
km altitude are seasonally averaged for each altitude bin and each subregion, and the
resulting vertical profiles of $\Delta CO_2$ are compared between simulations and observations.
**2.4 Metrics**
In order to evaluate the model performance to represent observations at different time scales
(annual, seasonal, synoptic, diurnal), following Cadule et al. (2010), we define a series of
metrics and corresponding statistics for each time scale. All the metrics, defined below, are
calculated for both observed and simulated $CH_4$ ($CO_2$) time series between 2006 and 2013.
2.4.1 Annual gradients between stations
As inversions use gradients to optimize surface fluxes, it is important to have a metric based
upon cross-site gradients. We take Hanle in India (HLE – 78.96°N, 32.78°E, 4517 m a.s.l.,
Figure 1, Table 2) as a reference and calculate the mean annual gradients by subtracting $CH_4$
($CO_2$) at HLE from those of other stations. HLE is a remote station in the free troposphere
within SEA and is located far from any important source/sink areas for both $CH_4$ and $CO_2$.
These characteristics make HLE an appropriate reference to calculate the gradients between
stations. Concentration gradients to HLE are calculated for both observations and model
simulations using the corresponding smoothed curves fitted with the CCGvu routine (see
Section 2.3). The ability of ZASIA and REG to represent the observed $CH_4$ ($CO_2$) annual
gradients across all the available stations is quantified by the mean bias (MB, Eq. 1) and the
root-mean-square deviation (RMSE, Eq. 2). In Eq. 1 and Eq. 2, $m_i$ and $o_i$ indicate





respectively the modelled and observed $CH_4$ ($CO_2$) mean annual gradient relative to HLE for
a station $i$.

$$MB = \frac{\sum_{i=1}^{N}(m_i - o_i)}{N} \qquad (1)$$


$$RMSE = \sqrt{\frac{\sum_{i=1}^{N}(m_i - o_i)^2}{N}} \qquad (2)$$


2.4.2 Seasonal cycle
Two metrics of the model ability to reproduce the observed $CH_4$ ($CO_2$) seasonal cycle are
considered, the phase and the amplitude. For each station, the seasonal phase is evaluated by
the Pearson correlation between the observed and simulated harmonics extracted from the
original time series, whereas the seasonal cycle amplitude is evaluated by the ratio of the
modelled to the observed seasonal peak-to-peak amplitudes based on the harmonics ($^{A_m}/_{A_o}$).
2.4.3 Synoptic variability
For each station, the performance of ZASIA and REG to represent the phase (timing) of the
synoptic variability is evaluated by the Pearson correlation coefficient between the modelled
and observed synoptic deviations (residuals) around the corresponding smoothed fitting curve
(see Section 2.3), whereas the performance for the amplitude of the synoptic variability is
quantified by the ratio of standard deviations of the residual concentration variability between
the model and observations (i.e., Normalized Standard Deviation, NSD, Eq. 3). Further, the
overall ability of a model to represent the synoptic variability of $CH_4$ ($CO_2$) at a station is
quantified by the RMSE (Eq. 4), a metric that can be represented with the Pearson correlation
and the NSD in a Taylor diagram (Taylor, 2001). In Eq. 3 and Eq. 4, $m_j$ ($o_j$) indicates the
modelled (observed) synoptic event $j$, whereas $\bar{m}$ ($\bar{o}$) indicates the arithmetic mean of all the
modelled (observed) synoptic events over the study period. Note that for the flask
measurements, $j$ corresponds to the time when a flask sample was taken, whereas for the
continuous measurements, $j$ corresponds to the early morning (00:00–03:00LST, for
mountain stations) or afternoon (12:00–15:00LST, for coastal or island stations) period of
each sampling day.




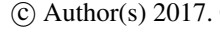

$$NSD = \frac{\sqrt{\frac{\Sigma_{j=1}^{N}(m_j - \overline{m})^2}{N}}}{\sqrt{\frac{\Sigma_{j=1}^{N}(o_j - \overline{o})^2}{N}}}$$
(3)

$$RMSE = \sqrt{\frac{\Sigma_{j=1}^{N}(m_j - o_j)^2}{N}}$$
(4)

2.4.4 Diurnal cycle
For each station, the model's ability to reproduce the mean $CH_4$ ($CO_2$) diurnal cycle phase in
a month is evaluated by the correlation of the hourly mean composite modeled and observed
values. With respect to the diurnal cycle amplitude, the model performance strongly varies
with stations and seasons, and we do not use a specific metric to evaluate it.
**3 Results and discussions**
**3.1 Annual gradients**
3.1.1 $CH_4$ annual gradients
The annual mean gradient between a station and the HLE reference station relates to the time
integral of transport of sources/sinks within the regional footprint area of the station on top of
the background gradient caused by remote sources. For $CH_4$, Figure 2a,b shows the
scatterplot of the simulated and observed mean annual gradients to HLE for all stations. In
general, both REG and ZASIA capture the signs of the observed $CH_4$ gradients with
reference to HLE, and the simulated gradients roughly distribute around the identity line
(Figure 2a,b). The linear regression slope of the simulated versus observed gradients is
$1.05\pm0.08$ for ZASIA (p<0.001, $R^2$=0.88) against $1.15\pm0.10$ (p<0.001, $R^2$=0.85) for REG.
The capability of ZASIA in representing the $CH_4$ gradients is thus slightly better than REG
within the zoomed region, with the mean bias and RMSE of -1.2±13.9 and 13.6 ppb for
ZASIA, compared to -5.4±20.1 and 20.3 ppb for REG (Table S1; Figure 2a,b). A better
performance of ZASIA compared to REG for the $CH_4$ gradients within SEA is also found for
all seasons (Figure S3), with the gradients in summer (AMJ and JAS) being generally better
captured than those in winter (OND and JFM). Outside the zoomed region, ZASIA does not
perform worse than REG, despite its degraded resolution.
When looking into the model performance for different station types, we note that ZASIA
performs better for reproducing the gradients vs. HLE at most marine, coastal and continental
stations within the zoom region, with a substantial reduction of RMSE compared to that
derived from REG (Table S1). In particular, with ZASIA, an improvement of the model
performance on the $CH_4$ annual gradient is found at Shangdianzi (SDZ – 117.12°E, 40.65°N,
293m a.s.l.) and Pondicherry (PON – 79.86°E, 12.01°N, 30m a.s.l.) (for flask measurements),
followed by Cape Rama (CRI – 73.83°E, 15.08°N, 66m a.s.l.) (Figure 2a,b), each having an
average bias reduction of 36.1 (98.6%), 30.5 (89.9%), 10.6 (74.2%) ppb respectively
compared to REG. This improvement may result from a reduction in representation error with
a higher model horizontal resolution in the zoomed region, through a better description of the
surface fluxes and/or transport around these stations. Particularly, given the presence of large
$CH_4$ emission hotspots in the surface emission maps (Figure S4), ZASIA makes the simulated
$CH_4$ fields more heterogeneous around emission hotspots with finer model grids (e.g., North
China in Figure S5), having the potential to better represent stations nearby on an annual
basis if the surface fluxes are prescribed with sufficient accuracy (see Figure S6 for SDZ).
Besides, stations located in complex terrains (e.g. coastal stations) are more likely to be well
characterized with a higher horizontal resolution, as shown in Figure S6 for PON and CRI.
However, a finer grid may also enhance model-data misfits related to inaccurate
meteorological forcings and/or surface flux maps. For example, for the coastal station Tae-
ahn Peninsula in Korea (TAP – 126.13°E, 36.73°N, 21m a.s.l.), both ZASIA and REG
overestimate the observed $CH_4$ gradients vs. HLE by more than +15 ppb, and ZASIA does
worse than REG (Figure S6). The overall poor model performance at this station suggests
that emission sources in the prescribed surface flux map are probably overestimated nearby
(also see the marine station GSN, Figure S6), although the uncertainty of OH distribution
could also play a role. Furthermore, as the $CH_4$ fields simulated with ZASIA are very
sensitive to large emission hotspots (Figure S5), representation of stations near hotspots
highly depends on the accuracy of the hotspot location and intensity in the surface emission
maps. For Ulaan Uul in Mongolia (UUM – 111.10°E, 44.45°N, 1012m a.s.l.), the existence of
the large hotspot (a coal mine) nearby in the EDGARv4.2FT2010 is uncertain, as it is not
represented in other inventories like the Regional Emission inventory in ASia (REASv2.1;
Kurokawa et al., 2013) (Figure S4). We find that ZASIA does not improve the representation
of the observed $CH_4$ gradients between UUM and HLE (Figure S6), probably due to the



presence of this (uncertain) hotspot in the prescribed surface emission maps. Besides, as
stated in several previous studies looking at the variability of tracers over continental areas
(Geels et al., 2007; Law et al., 2008; Patra et al., 2008), for a station located in a complex
terrain (e.g. coastal or mountain sites), the selection of an appropriate gridpoint and/or model
level to represent an observation is challenging. In this study we sample the gridpoint and
model level nearest to the location of the station, which may not be the best representation of
flask data sampling selection strategy (e.g. marine sector at coastal stations or strong winds)
and could contribute to the model-observation misfits.
3.1.2 $CO_2$ annual gradients
For $CO_2$, both ZASIA and REG are generally able to capture the mean annual gradients
between stations, although not as well as for $CH_4$ (Figure 2c,d). The linear regression slopes
between the simulated and the observed gradients are $0.63\pm0.10$ ($p<0.001$, $R^2=0.56$) and
$0.65\pm0.10$ ($p<0.001$, $R^2=0.59$) for ZASIA and REG, respectively. Note that both ZASIA and
REG are not able to reproduce the $CO_2$ gradients at tropical stations like Bukit Kototabang in
Western Indonesia (BKT – 100.32°E, 0.20°S, 869m a.s.l.), Lulin in Taiwan (LLN – 120.87°E,
23.47°N, 2867m a.s.l.), and SNG and CRI in Western India, with mean biases (absolute value)
ranging between 2.5–6.3 ppm, compared to 0.0–1.9 ppm at other sites within the zoomed
region (except TAP, see details below). For BKT, the model-observation discrepancies are
probably due to imperfect NEE fluxes and/or fire emissions in the prescribed surface fluxes.
For SNG and CRI, the significant negative biases of models found during the Northeast
monsoon season (October–March) may suggest an underestimation of $CO_2$ sources in the
upwind regions (Figure S7e-h). Figure 2 also shows that, in contrast with $CH_4$, ZASIA does
not significantly improve the representation of $CO_2$ gradients for stations within the zoomed
region, with the mean bias and RMSE close to those of REG. At TAP, ZASIA even degrades
model performance (Figure S8), possibly related to misrepresentation of $CO_2$ sources in the
prescribed surface flux map and transport effects. If the ZASIA output is sampled one grid
offshore from TAP, the modelled $CO_2$ gradients become more consistent with observations
(Figure S8).
3.1.3 $CH_4$ vs $CO_2$
With ZASIA, the model improvement to represent the GHG gradients is more apparent for
$CH_4$ than for $CO_2$. This difference points towards the quality of source fields for $CO_2$, and

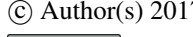



especially natural ones. They are spatially more diffuse than those of $CH_4$, and temporally
more variable in response to weather changes (Parazoo et al., 2008; Wang et al., 2007).
Therefore, the regional differences of NEE in SEA not captured by the global ORCHIDEE
model may explain the worse fit of the $CO_2$ gradients compared to $CH_4$ in both ZASIA and
REG. Further, the spatial resolution of the prescribed flux maps may also account for the
different model performance for $CO_2$ and $CH_4$ (e.g. the spatial resolution of the
anthropogenic emissions is 1° for $CO_2$ and 0.1° for $CH_4$ respectively). Therefore, with our
setup of surface fluxes (Table 1), ZASIA is more likely to resolve the spatial heterogeneity of
$CH_4$ fields, and its improvement over REG is more apparent than that for $CO_2$. Note that for
$CO_2$, simulations have been performed with the fossil fuel and NEE fluxes that are variable at
monthly, daily and hourly scales, respectively. Results show that accounting for daily or
hourly variability of fossil $CO_2$ emissions and NEE fluxes produces almost the same mean
annual $CO_2$ gradients as the simulations prescribed with monthly constant fluxes in each
emitting grid-cell. This indicates that the synoptic and diurnal variations of the fossil fuel
emissions and NEE from the current ORCHIDEE outputs do not have strong impacts on
representation of the mean annual gradients. In the rest of this paper we only present the
results driven by hourly fossil fuel emissions and NEE.
**3.2 Seasonal cycles**
3.2.1 $CH_4$ seasonal cycles
The model performance for the seasonal cycle depends on the quality of seasonal surface
fluxes, atmospheric transport, and chemistry (for $CH_4$ only). For $CH_4$, as shown in Figure 3a,
both ZASIA and REG generally capture the $CH_4$ seasonal phases, with correlation
coefficients larger than 0.8 at 19 out of 26 stations (73%) for both model verions. However,
they are less capable to represent the seasonal cycle at mountain stations, such as Plateau
Assy in Kazakhstan (KZM – 77.87°E, 43.25°N, 2524m a.s.l.) (Pearson correlation R < 0, not
shown in Figure 3a) and Waliguan (WLG – 100.90°E, 36.28°N, 3890m a.s.l.) (Pearson
correlation R < 0.4, Figure 3a). Compared to REG, ZASIA seems to perform better at
elevated stations (see WLG, UUM and probably also SNG in Figure 3a). On the other hand,
for stations where the $CH_4$ seasonal phases are already well simulated, ZASIA does not
significantly improve the model performance over REG. For stations ouside the zoomed



region (e.g., GMI, COI, RYO), the performance of REG is better given the degraded model
resolution.
With respect to the $CH_4$ seasonal amplitude, for 10 out of 26 stations (inside the dotted circle
in Figure 3b), the amplitude ratio $A_m/A_o$ is within the range 0.75–1.25 for both ZASIA and
REG. Among the four station types (symbols with different colors in Figure 3b), the seasonal
amplitudes of marine stations are well represented by both model versions, while the seasonal
amplitudes at continental (SDZ, WIS, KZD) and mountain stations (e.g. SNG, KZM) are
more difficult to capture. Note that for the stations with the seasonal amplitude ratios $A_m/A_o$
outside the range 0.75–1.25, ZASIA substantially improves the model performance compared
to REG (see stations in the shaded areas in Figure 3b).
At mountain stations, the $CH_4$ seasonal cycle is less accurately represented than for other
station types in both model versions. However, ZASIA moderately improves the model
performance in several cases, probably through better resolved topography, surface fluxes
and/or horizontal transport. For example, the $CH_4$ seasonal amplitude is better simulated at
HLE by ZASIA, in particular for the summer maximum (Figure S9). Given that HLE is a
mid-tropospheric background station with negligible influence from local sources (Lin et al.,
2015), the improvement by ZASIA suggests that a finer horizontal grid allows for a better
representation of the seasonal transport impacting $CH_4$ at HLE. However, at WLG and KZM,
the improvement of the transport with ZASIA is very limited (Figure S9). Possible reasons
that account for the model-observation mismatch at mountain sites could be related to the
model sampling strategy, and the accuracy of surface flux maps (e.g. unresolved $CH_4$ sources
in summer or possible emission hotspots near stations) and OH distribution. Moreover, the
vertical resolution of both model versions is rather coarse (19 layers), and recent tests with
the updated physical parameterizations and increased vertical resolution have shown
significant improvement of the model performance on vertical transport (Locatelli et al.,

503    2015).

We also note that the $CH_4$ seasonal cycle at several continental and coastal stations (e.g., SDZ,
TAP and GSN; Figure S9) are also not well captured by both model versions. At SDZ in
North China, the $CH_4$ seasonal amplitude is overestimated by 66 ppb (139%) and by 45 ppb





(94%) in ZASIA and REG respectively, with an overestimated summer maximum (Figure
S9). Given that SDZ is influenced by air masses passing over several megacities in North
China (Beijing, Tianjin, and Tangshan) in summer (Fang et al., 2016), the overestimation of
the summer maximum suggests that the actual $CH_4$ emissions from this region should be
lower than the prescribed values in our simulations, apart from the possible influence of
inaccurate OH distribution. In addition, the $CH_4$ seasonal amplitudes at TAP and GSN in the
Korean Pennisula are also overestimated by both model versions, suggesting that the
prescribed $CH_4$ emissions may be overestimated in East Asia as well (particularly China).
This is consistent with the analyses of the annual $CH_4$ gradients (Section 3.1), and further
reinforced by results from other independent inventories (e.g., Peng et al., 2016) and inverse
modeling (Bergamaschi et al., 2013; Bruhwiler et al., 2014; Thompson et al., 2015).
3.2.2 $CO_2$ seasonal cycles
The $CO_2$ seasonal cycle mainly represents the seasonal cycle of NEE from ORCHIDEE
convoluted with atmospheric transport. Figure 3c illustrates that both ZASIA and REG are
able to capture the $CO_2$ seasonal phases at most stations. A high correlation (Pearson
correlation R > 0.8) is found between the simulated and observed $CO_2$ harmonics for 25 out
of 31 stations. ZASIA does not significantly improve the model performance for most
stations where the $CO_2$ phase is already well represented by REG. We note that neither model
versions captures the $CO_2$ seasonal phase at BKT in western Indonesia, no matter whether the
evaluation is performed against flask or continuous measurements. Given that representation
of the $CH_4$ seasonal cycle at BKT was satisfactory (Figure S9 for analyses of flask
measurements), the worse model performance for $CO_2$ suggests inaccurate prescribed surface
fluxes for NEE and/or fire emissions.
With respect to the $CO_2$ seasonal amplitude, 14 out of 31 stations have the amplitude ratios
$A_m/A_o$ ranging 0.75–1.25 from both ZASIA and REG (symbols inside the dotted circle in
Figure 3d). For the other stations, both model versions tend to underestimate the $CO_2$
amplitudes, and ZASIA does not improve the model performance. As for $CH_4$, both ZASIA
and REG do not well capture the $CO_2$ amplitudes at mountain stations (e.g., SNG, LLN and
HLE) and continental stations (e.g. SDZ) compared to other station types (e.g. Figure S10). A
likely cause can be the inaccurate estimation of NEE in ORCHIDEE (e.g., Peng et al., 2015).





For mountain stations, transport errors and the model sampling strategy may additionally
account for the model-observation discrepancies. As mentioned for the annual gradients
(Section 3.1), simulations prescribed with monthly, daily and hourly $CO_2$ surface fluxes
generate nearly the same $CO_2$ seasonal cycle for each station, indicating that the seasonal
changes in the magnitude of the $CO_2$ diurnal rectification do not significantly modulate the
average seasonal cycle.
**3.3 Synoptic variability**
3.3.1 $CH_4$ synoptic variability
The day-to-day variability of $CH_4$ and $CO_2$ residuals are influenced by the regional
distribution of fluxes and atmospheric transport at the synoptic scale. For $CH_4$, as shown in
Figure 4a,b, both ZASIA and REG generally capture the synoptic variability at most stations.
Apart from UUM where the model performance is poor possibly due to the wrong hotspot
near the site in EDGARv4.2 FT2010 (see UUM in Figure S6), for ZASIA (REG), 63% (59%)
stations have correlation coefficients (R) higher than 0.5. Both model versions give an NSD
range of 0.5–2.0; at stations where the amplitude is not well captured by REG, ZASIA tends
to improve the model performance with NSDs closer to 1 (e.g. stations TAP, WLG, SNG,
PON in Figure 4a,b).
Among the four station types, the synoptic variability at marine and coastal stations is better
simulated, especially within the zoomed region (Figure 4a,b). Given that the prescribed $CH_4$
surface fluxes are monthly averages, the overall good model performance at marine and
coastal stations suggests that variations in atmospheric transport account for most of the $CH_4$
synoptic variability at these stations. Two exceptions are GSN and TAP (Figure 4a,b). Both
stations are not adequately represented, with the amplitudes overestimated throughout the
year, by a factor of 1.2–2.3 (Figure S11). These results, together with the overestimated $CH_4$
gradients (to HLE) and seasonal amplitudes presented before (see Section 3.1 and Section
3.2), again suggest that the prescribed surface fluxes are overestimated near these two
stations. The synoptic variability at mountain stations is also fairly well represented by both
model versions except WLG and SNG. At SNG, both the phase and amplitude are not well
captured, especially in winter. By contrast, the coastal station CRI nearby is much better
represented (Figure 4a,b, Figure S12). Given that the two stations may be influenced by air
masses from similar source regions, the representation of vertical transport could be



problematic at SNG. Compared to other station types, the synoptic variability at continental
stations is more difficult to capture (e.g. SDZ and KZD in Figure 4a,b, and UUM not shown),
possibly because of errors in the prescribed surface fluxes or/and error in transport over
complex continental terrains. Note that at several stations (as shown in Figure S12 for SNG,
SDZ and UUM), there are a few synoptic events that are not realistically simulated with very
large model-observation misfits. Caution should be taken when assimilating observations
from these stations.
3.3.2 $CO_2$ synoptic variability
For $CO_2$, as shown in Figure 4c,d, representation of the synoptic variability is overall not as
good as for $CH_4$. Based on model outputs from ZASIA (REG), only 10% (13%) stations have
correlation coefficients (R) higher than 0.5, and the NSD values range 0.3–1.4 (0.3–6.3).
ZASIA does not significantly improve over REG. This is also true when one looks into
different station types or different seasons (Figure S13). At several tropical stations where the
$CH_4$ synoptic variability is found to be simulated correctly (e.g., BKT and PON, Figure 4,
Figure S14), both ZASIA and REG are not able to reproduce the phase of the $CO_2$ synoptic
variability. Given that the transport process is the same for both gases in the model, the
overall degraded model performance for $CO_2$ compared to $CH_4$ suggests biases in the NEE
flux fields from ORCHIDEE. As noted by several previous studies (e.g., Patra et al., 2008),
$CO_2$ fluxes with sufficient accuracy and resolution are indispensable for representing the $CO_2$
synoptic variability. In this study, the daily to hourly NEE variability does not seem to be
well represented in ORCHIDEE, especially in the tropics, hence both ZASIA and REG do
not capture the $CO_2$ synoptic variability at stations strongly influenced by regional land
fluxes. Futher, at stations influenced by episodic fire emissions (e.g., BKT, and probably
PON and PBL), the monthly averaged biomass burning emissions used in this study may not
realistically simulate the $CO_2$ synoptic variability due to the coarse temporal resolution.
Besides, the resolution of the prescribed $CO_2$ ocean fluxes are also rather coarse, which may
additionally account for the relatively poor model performance on the $CO_2$ synoptic
variability compared to $CH_4$, especially for marine and coastal stations.



### 3.4 Diurnal cycle


3.4.1 $CH_4$ diurnal cycle
The diurnal cycles of trace gases are mainly controlled by the co-variations between local
fluxes and boundary layer mixing (i.g. the diurnal rectifier effect; Denning et al., 1996). For
$CH_4$, the correlation between the simulated and observed diurnal cycles ranges 0.19–0.64 for
ZASIA, and the model performance is generally better than that of REG, especially at
stations within the zoomed region (Table S2). However, the values of the correlation
coefficients remain low. Apart from AMY and COI where both ZASIA and REG fairly well
capture the $CH_4$ diurnal phase, the model performance at other stations is poor and varies
from month to month (Table S2). The same conclusion was reached in a previous study by
Patra et al. (2009b), which used the ATCM chemistry transport model to simulate $CH_4$
concentrations at a horizontal resolution of 2.8°×2.8°. Compared to ATCM in Patra et al.
(2009b), ZASIA seems to better represent the $CH_4$ diurnal cycle at coastal stations (AMY,
COI, GSN, HAT, MNM and RYO used in both studies). This improved model performance
may be at least partly attributed to a better representation of the coastal topograhy and diurnal
changes in the land-sea breeze with the finer horizontal grids.
Here, increasing the horizontal resolution does not improve representation of the $CH_4$ diurnal
cycles, therefore several other aspects may account for the model-observation mismatch. First,
the prescribed surface fluxes averaged monthly may not be adequate to resolve the diurnal
variations at stations strongly influenced by local and regional sources. For example, as
shown in Figure 5a,b, the $CH_4$ diurnal amplitude at AMY is fairly well simulated in winter
but substantially underestimated by both ZASIA and REG in summer. Given that emissions
from wetlands and rice paddies nearby should affect its $CH_4$ signals in summer, and that these
emissions are temperature and moisture dependent, the climatological surface fluxes used in
this study may not be able to reproduce the $CH_4$ diurnal cycles. Second, model performance
also relies on representation of the diurnal variations of boundary layer mixing in the model.
The coarse vertical resolution of the transport model (19 levels) has recently been proved to
limit the model's ability to account for various dynamical and physical processes in the
planetary boundary layer (Locatelli et al., 2015). Locatelli et al. (2015) showed that the
refined vertical resolution, together with the updated physical parameterization on turbulent
diffusion and convection, can substantially improve performance of the LMDz model on the



diurnal cycle of trace gases. Furthermore, representation of the $CH_4$ diurnal cycle is even
more challenging for mountain stations affected by daytime upslope winds and nighttime
subsidence (Griffiths et al., 2014; Pérez-Landa et al., 2007). At BKT, located on an altitude
of 869 m a.s.l., both model versions do not capture the $CH_4$ diurnal cycle well when models
are sampled at this altitude. However, the first level of the model show better agreement with
the observed diurnal cycle at BKT than the level of the station. (Figure 5c,d). This suggests
that the vertical transport around this station may not be well represented by both model
versions, including the upslope winds that connect the surface layer to the station.
3.4.2 $CO_2$ diurnal cycle
For $CO_2$, the correlation between the simulated and observed diurnal cycle ranges from -0.34
to 0.78 for ZASIA. Compared to REG, ZASIA does not always perform better for stations in
the zoomed region; in some instances, its performance is even worse (Table S3). As already
mentioned, we believe that this poor performance is due to biases in the diurnal cycle of NEE
from ORCHIDEE coupled to biases in the diurnal cycle of vertical transport. For example, at
GSN (Figure 6a,b), both ZASIA and REG well capture the timing and amplitude of the $CO_2$
diurnal cycle in winter but not in summer, underlining importance of the data quality of land
surface $CO_2$ fluxes on the model performance, especially during periods when biospheric
uptake and release are active. For stations strongly influenced by local sources, the amplitude
of the $CO_2$ diurnal cycle tends to be underestimated. This is the case for PON (Figure 6c,d)
on the southeast coast of India, 8 km north of the city of Pondicherry with a population of
more than 240,000 in 2011 (Lin et al., 2015). On the one hand, both ZASIA and REG do not
adequately capture the diurnal rectifier effect (Denning et al., 1996), since the diurnal cycle
of the prescribed NEE fluxes is underestimated and the boundary layer mixing is not well
represented by the current model (Locatelli et al., 2015). On the other hand, given that local
sources are unresolved in the prescribed $CO_2$ emissions at 1°×1° resolution, and that the
boundary layer air is not well-mixed during nighttime, neither model version captures the
$CO_2$ daily maxima. Again at BKT, as noted for the $CH_4$ diurnal cycle, we also find better
model-observation agreement for the $CO_2$ diurnal cycle when sampling the first model layer
rather than the one corresponding to the station height (Figure 6e,f). Overall, ZASIA does not
significantly improve the representation of the $CO_2$ diurnal cycle (Table S3), even when
hourly $CO_2$ fossil fuel and NEE fluxes are prescribed.





### 3.5 Evaluation against the CONTRAIL CO$_2$ vertical profiles

Figure 7 shows the simulated and observed CO$_2$ vertical profiles averaged for different seasons and over different regions. Over East Asia (EAS; Figure 7a and Figure S1), both ZASIA and REG reasonably reproduce the shape of the observed CO$_2$ vertical profiles. During April–June (AMJ), despite a well-simulated CO$_2$ vertical profile, the modelled $\Delta$CO$_2$ (see Section 2.3 for details) are consistently lower than the observations by about 2–3 ppm throughout all altitude bins, possibly due to earlier spring CO$_2$ uptake simulated by ORCHIDEE in East Asia (also see the CO$_2$ seasonal cycle at GSN and KIS in Figure S10). The simulated CO$_2$ vertical gradients between planetary boundary layer (BL) and free troposphere (FT) are lower than the observations by 1–2 ppm during the winter seasons (Figure 8a), possibly due to stronger vertical mixing in LMDz (Locatelli et al., 2015; Patra et al., 2011) as well as flux uncertainty. Note that as most samples (79%) are taken over the Narita International Airport (NRT) and Chubu Centrair International Airport (NGO) in Japan, both of which are located outside the zoomed region, REG better reproduces the BL-FT gradients than ZASIA.

Over the Indian sub-continent (IND, Figure 7b), there is large underestimation of the magnitude of $\Delta$CO$_2$ near the surface by 4–5 ppm during AMJ, July–September (JAS) and October–December (OND). Accordingly, the BL-FT gradient is underestimated by the same magnitude for these periods (Figure 8b). The model-observation discrepancies are probably not only related to uncertainties in the vertical mixing processes, but also to imperfect surface fluxes throughout the development of the Indian summer monsoon system. This result is consistent with the model underestimation of the CO$_2$ seasonal amplitude at most surface stations in this region (Figure 3d, Figure S10). When considering the CH$_4$ and CO$_2$ measurements at the two mountain station HLE (4517 m a.s.l.) and SNG (1600 m a.s.l.) and a coastal station CRI (66 m a.s.l.) in India, both ZASIA and REG well capture the phase of the CH$_4$ seasonal cycle at all three stations (Figure S9), whereas the CO$_2$ seasonal phase is not so well simulated, especially at HLE and SNG (Figure S10). The simulated seasonal CO$_2$ maximum and minimum at the two inland mountain stations are earlier than the observed ones by up to 1–2 months. This implies that the prescribed NEE does not adequately capture the phenology as well as the magnitude of the strong sources and sinks during the pre-monsoon and monsoon seasons over the Indian sub-continent (Valsala et al., 2013), although





the vertical transport (including deep convection) over the periods may also not be well
represented in the LMDz model.
The $CO_2$ vertcial profiles over Southeast Asia (including Northern Southeast Asia (NSA) and
Southern Southeast Asia (SSA)) are generally well reproduced. However, both ZASIA and
REG fail to reproduce the BL-FT gradient of 3–4 ppm in April for NSA, and during August–
October for SSA (Figure 8c,d). Apart from errors in the vertical transport and prescribed NEE,
inaccurate estimates of biomass burning emissions could also contribute to this model-
observation mismatch.
Overall, the $CO_2$ vertical profiles from the CONTRAIL project are fairly well simulated by
ZASIA and REG over SEA, despite underestimation of the BL-FT gradients, particularly
over the Indian sub-continent. This model-observation mismatch are due to a mix of
imperfect representation of both vertical transport and surface fluxes, and can not be
significantly reduced by solely refining the horizontal resolution of the model, as shown by
the very similar vertical profiles derived from ZASIA and REG for $CO_2$. In order to improve
the model performance on the vertical profiles of trace gases (especially the gradients near
the surface), the vertical resolution of the model should be increased, together with
implementation of the updated physical parameterization for e.g. boundary layer mixing and
deep convection in the troposphere (Locatelli et al., 2015).
**4 Conclusions and implications**
We have simulated the 4-D concentration fields of $CH_4$ and $CO_2$ over South and East Asia
(SEA) using a zoomed chemical transport model ZASIA. We have evaluated the model's
ability to simulate the $CH_4$ and $CO_2$ variability at multi-annual, seasonal, synoptic and diurnal
scales, against flask and continuous measurements from a unique dataset of 30 surface
stations. To assess the model performance, $CH_4$ and $CO_2$ are also simulated using the same
chemical transport model without the zoom (REG). The results show that both ZASIA and
REG are generally capable of representing the annual gradients and seasonal cycles of $CH_4$
and $CO_2$, with overall better model performance for $CH_4$ than $CO_2$. Compared to REG,
ZASIA moderately improves representation of the $CH_4$ gradients and seasonal cycle; for $CO_2$,
the performance of the two model versions do not show a significant difference, suggesting
issues with the surface fluxes used. At the synoptic scale, the $CH_4$ variability is captured





fairly well for most stations, especially marine and coastal stations, while representation of
the $CO_2$ synoptic variability is not as good as $CH_4$ even when high-frequency (hourly)
anthropogenic emissions and NEE fluxes are prescribed. The model's ability to reproduce the
$CH_4$ and $CO_2$ diurnal cycle is relatively poor except for a few stations, and better
representation of the diurnal cycles cannot be achieved solely with a higher horizontal
resolution and with the current model setups. The evaluation at different temporal scales and
comparisons between different species and model horizontal resolutions have given us
information on possible model improvements needed and implications for inverse modeling,
which we summarize in the following paragraphs.
First, the performance of the zoomed chemical transport model is moderately better than
REG for $CH_4$ in SEA. The $CH_4$ measurements from regional stations and high-frequency
sampling are generally better represented with a finer horizontal resolution. This improved
representation of the $CH_4$ variability integrates better description of the topography, the
transport and/or the $CH_4$ surface fluxes around stations. Particularly, given the existence of
large $CH_4$ emission hotspots over SEA, the zoomed transport model simulates more
heterogeneous $CH_4$ fields around emission hotspots and improves representation of stations
nearby. However, the strong sensitivity of the simulated $CH_4$ to emission hotspots also means
that the performance of the zoomed transport model depends more on the accuracy of the
location and the magnitude of emission hotspots in the prescribed surface fluxes than the
regular transport model. As representation of emission hotspots are uncertain in the current
bottom-up inventories, caution should be taken when one assimilates observations from
stations nearby, particularly those that are unrealistically simulated by the transport model
(e.g. the cases for the synoptic variability at SDZ and UUM).
Second, the lower model performance for $CO_2$ compared to $CH_4$ at all temporal scales
suggests that the $CO_2$ surface fluxes have not been prescribed with sufficient accuracy. This
is particularly true for stations in South and Southeast Asia where NEE does not seem to be
well simulated by the terrestrial ecosystem model ORCHIDEE. Given that the bottom-up
estimates of $CO_2$ fluxes from emission inventories and ecosystem models often suffer from
large uncertainties, high resolution inverse modeling of $CO_2$ fluxes could help optimally
combine information from atmospheric measurements to improve our knowledge of $CO_2$
fluxes and variabilities.





Third, the performance of the zoomed transport model should be further enhanced by refining
the vertical resolution as well as by improving representation of the vertical transport. With
the current setups of model layers and physical parameterizations, improvement of model
performance is not apparent on hourly to weekly timescales. In addition to improving data
quality of the prescribed surface fluxes, the low vertical resolution also limits the models'
ability to represent the $CH_4$ and $CO_2$ variability at short timescales. In order to take advantage
of high-frequency observations at stations close to source regions, it is recommended to
increase the model vertical resolution and to improve representation of boundary layer
mixing. Such efforts are ongoing with LMDz but will be implemented soon.
Lastly, the model-observation comparisons at multiple temporal scales have the potential to
inform us about the magnitude of sources and sinks in the studied region. For example, at
GSN, TAP and SDZ, all of which located in East and Northeast Asia, the $CH_4$ annual
gradients as well as the amplitudes of seasonal and synoptic variability are consistently
overestimated, suggesting that the prescribed $CH_4$ emissions in East Asia are overestimated.
Atmospheric inversions that assimilate information from these stations are expected to
decrease emissions in East Asia, which we will further investigate in future inversion studies.



**Acknowledgement**
This study was initiated within the framework of CaFICA-CEFIPRA project (2809-1). X. Lin
acknowledges the PhD funding support from AIRBUS Defense & Space. P. Ciais thanks the
ERC SyG project IMBALANCE-P 'Effects of Phosphorus Limitations on Life, Earth System
and Society' Grant agreement (no. 610028). N. Evangeliou acknowledges the Nordic Center
of Excellence eSTICC project (eScience Tools for Investigating Climate Change in northern
high latitudes) funded by Nordforsk (no. 57001). We acknowledge the WDCGG for
providing the archives of surface station observations for $CO_2$ and $CH_4$. We thank the
following networks or institutes for the efforts on surface GHG measurements and their
access: NOAA/ESRL, Aichi, BMKG, CMA, CSIR4PI, CSIRO, Empa, ESSO/NIOT, IIA,
IITM, JMA, KMA, LSCE, NIER, NIES, PU and Saitama. Finally, we would like to thank F.
Marabelle and his team at LSCE, and the CURIE (TGCC) platform for the computing
support.





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

**Tables**
**Table 1** The prescribed $CH_4$ and $CO_2$ surface fluxes used as model input. For each trace gas,
magnitudes of different types of fluxes are given for the year 2010. $Total_{global}$ and $Total_{zoom}$
indicate the total flux summarized over the globe and the zoomed region, respectively.

| Type of $CH_4$ fluxes | Temporal resolution | Spatial resolution | $Total_{global}$ ($TgCH_4$/yr) | $Total_{zoom}$ ($TgCH_4$/yr) | Data source |
|---|---|---|---|---|---|
| Anthropogenic – rice | Monthly, interannual | 0.1° | 38 | 32 | EDGARv4.2FT2010 + Matthews et al (1991) |
| Anthropogenic – others | Yearly, interannual | 0.1° | 320 | 131 | EDGARv4.2FT2010 |
| Wetland | Monthly, climatological | 1° | 175 | 29 | Kaplan et al. (2006) |
| Biomass burning | Monthly, interannual | 0.5° | 19 | 3 | GFED v3.0 |
| Termite | Monthly, climatological | 1° | 19 | 3 | Sanderson et al. (1996) |
| Soil | Monthly, climatological | 1° | -38 | -7 | Ridgwell et al. (1999) |
| Ocean | Monthly, climatological | 1° | 17 | 3 | Lambert & Schmidt (1993) |
| Total, $TgCH_4$/yr | | | 550 | 194 | |
| Type of $CO_2$ fluxes | Temporal resolution | Spatial resolution | $Total_{global}$ (PgC/yr) | $Total_{zoom}$ (PgC/yr) | Data source |
| Anthropogenic | Monthly, interannual | 1° | | | |
| Anthropogenic | Daily, interannual | 1° | 8.9 | 3.6 | IER-EDGAR product |
| Anthropogenic | Hourly, interannual | 1° | | | |
| Biomass burning | Monthly, interannual | 0.5° | 2.0 | 0.2 | GFED v3.1 |
| Land flux (NEE) | Monthly, interannual | 0.5° | | | |
| Land flux (NEE) | Daily, interannual | 0.5° | -2.7 | 0.1 | OCHIDEE outputs from trunk version r1882 |
| Land flux (NEE) | Hourly, interannual | 0.5° | | | |
| Ocean flux | Monthly, interannual | 4°×5° | -1.3 | 0.1 | NOAA/PMEL & AOML product; Park et al. (2010) |
| Total, PgC/yr | | | 6.9 | 3.9 | |


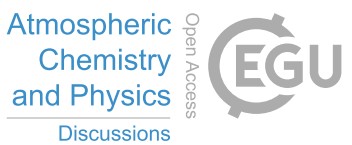

**Table 2** Stations used in this study. For the column 'Zoom', 'Y' indicates a station within the zoomed region.

| | Code | Station | LON (°) | LAT (°) | Altitude (m a.s.l.) | Contributor | Type | Temporal coverage | Zoom | CH$_4$ | CO$_2$ |
|---|---|---|---|---|---|---|---|---|---|---|---|
| 1 | AMY | Anmyeon-do, Korea | 126.32 | 36.53 | 133 | KMA | coastal | 2006–2013 | Y | Y | |
| 2 | BKT | Bukit Kototabang, Indonesia | 100.32 | -0.20 | 869 | BMKG, Empa, NOAA/RSRL | coastal | Flask: 2006–2013 CH$_4$ continuous: 2009–2013 CO$_2$ continuous: 2010–2013 | Y | Y | Y |
| 3 | COI | Cape Ochi-ishi, Japan | 145.50 | 43.16 | 94 | NIES | coastal | 2006–2013 | Y | Y | |
| 4 | CRI | Cape Rama, India | 73.83 | 15.08 | 66 | CSIRO | coastal | 2009–2013 | Y | Y | Y |
| 5 | DDR | Mt. Dodaira, Japan | 139.18 | 36.00 | 840 | Saitama | continental | 2006–2013 | Y | | Y |
| 6 | DSI | Dongsha Island, Taiwan, China | 116.73 | 20.70 | 8 | National Central Univ., NOAA/ESRL | marine | 2010–2013 | Y | Y | Y |
| 7 | GMI | Mariana Island, Guam | 144.66 | 13.39 | 5 | Univ. of Guam, NOAA/ESRL | marine | 2006–2013 | | Y | Y |
| 8 | GSN | Gosan, Korea | 126.12 | 33.15 | 144 | NIER | marine | 2006–2011 | Y | Y | Y |
| 9 | HAT | Hateruma, Japan | 123.81 | 24.06 | 47 | NIES | marine | 2006–2013 | Y | Y | Y |
| 10 | HLE | Hanle, India | 78.96 | 32.78 | 4517 | LSCE, CSIR4PI, IIA | mountain | Flask: 2006–2013 CH$_4$ continuous: 2012–2013 CO$_2$ continuous: 2006–2013 | Y | Y | Y |
| 11 | JIN | Jinsha, China | 114.20 | 29.63 | 750 | CMA | continental | 2006–2011 | Y | | Y |
| 12 | KIS | Kisai - Saitama | 139.55 | 36.08 | 13 | Saitama | continental | 2006–2013 | | | Y |
| 13 | KZD | Sary Taukum, Kazakhstan | 75.57 | 44.45 | 412 | KSIEMC, NOAA/ESRL | continental | 2006–2009 | Y | Y | Y |
| 14 | KZM | Plateau Assy, Kazakhstan | 77.87 | 43.25 | 2524 | KSIEMC, NOAA/ESRL | mountain | 2006–2009 | Y | Y | Y |
| 15 | LIN | Lin'an, China | 119.72 | 30.30 | 139 | CMA | continental | 2006–2011 | Y | | Y |
| 16 | LLN | Lulin, Taiwan, China | 120.87 | 23.47 | 2867 | LAIBS, NOAA/ESRL | mountain | 2006–2013 | Y | Y | Y |
| 17 | LON | Longfengshan, China | 127.60 | 44.73 | 331 | CMA | continental | 2006–2011 | Y | | Y |
| 18 | MKW | Mikawa-Ichinomiya, Japan | 137.43 | 34.85 | 50 | Aichi | continental | 2006–2011 | Y | | Y |
| 19 | MNM | Minamitori-shima, Japan | 153.98 | 24.28 | 28 | JMA | marine | 2006–2013 | | Y | Y |
| 20 | PBL | Port Blair, India | 92.76 | 11.65 | 20 | LSCE, CSIR4PI, ESSO/NIOT | marine | 2009–2013 | Y | Y | Y |
| 21 | PON | Pondicherry, India | 79.86 | 12.01 | 30 | LSCE, CSIR4PI, Pondicherry Univ. | coastal | Flask: 2006–2013 CH$_4$ continuous: 2011–2013 CO$_2$ continuous: 2011–2013 | Y | Y | Y |
| 22 | RYO | Ryori, Japan | 141.82 | 39.03 | 280 | JMA | continental | 2006–2013 | | Y | Y |
| 23 | SDZ | Shangdianzi, China | 117.12 | 40.65 | 293 | CMA, NOAA/ESRL | continental | 2009–2013 | Y | Y | Y |
| 24 | SEY | Mahe Island, Seychelles | 55.53 | -4.68 | 7 | SBS, NOAA/ESRL | marine | 2006–2013 | | Y | Y |
| 25 | SNG | Sinhagad, India | 73.75 | 18.35 | 1600 | IITM | mountain | CH$_4$ flask: 2010–2013 | Y | Y | Y |






| | | | | | | | | | CO$_2$ flask: 2009–2013 | | |
|---|---|---|---|---|---|---|---|---|---|---|---|
| 26 | TAP | Tae-ahn Peninsula, Korea | 126.13 | 36.73 | 21 | KCAER, NOAA/ESRL | coastal | 2006–2013 | | Y | Y |
| 27 | UUM | Ulaan Uul, Mongolia | 111.10 | 44.45 | 1012 | MHRI, NOAA/ESRL | continental | 2006–2013 | | Y | Y |
| 28 | WIS | Negev Desert, Israel | 30.86 | 34.79 | 482 | WIS, AIES, NOAA/ESRL | continental | 2006–2013 | Y | Y | Y |
| 29 | WLG | Mt. Waliguan, China | 100.90 | 36.28 | 3890 | CMA, NOAA/ESRL | mountain | 2006–2013 | Y | Y | Y |
| 30 | YON | Yonagunijima, Japan | 123.02 | 24.47 | 50 | JMA | marine | 2006–2013 | Y | Y | Y |

Abbreviations:

Aichi – Aichi Air Environment Division, Japan

AIES – Arava Institute for Environmental Studies, Israel

BMKG – Agency for Meteorology, Climatology and Geophysics, Indonesia

CMA – China Meteorological Administration, China

CSIR4PI – Council of Scientific and Industrial Research Fourth Paradigm Institute, India

CSIRO – Commonwealth Scientific and Industrial Research Organisation, Australia

Empa – Swiss Federal Laboratories for Materials Testing and Research, Switzerland

ESSO/NIOT – Earth System Sciences Organisation/National Institute of Ocean Technology, India

IIA – Indian Institute of Astrophysics, India

IITM – Indian Institute of Tropical Meteorology, India

JMA – Japan Meteorological Agency, Japan

KCAER – Korea Centre for Atmospheric Environment Research, Republic of Korea

KMA – Korea Meteorological Administration, Republic of Korea

KSIEMC – Kazakh Scientific Institute of Environmental Monitoring and Climate, Kazakhstan

LAIBS – Lulin Atmospheric Background Station, Taiwan

LSCE – Laboratoire des Sciences du Climat et de l'Environnement, France

MHRI – Mongolian Hydrometeorological Research Institute, Mongolia

NIER – National Institute of Environmental Research, South Korea

NIES – National Institute for Environmental Studies, Japan

NIWA – National Institute of Water and Atmospheric Research, New Zealand

NOAA/ESRL – National Oceanic and Atmospheric Administration/Earth System Research Laboratory

Saitama – Center for Environmental Science in Saitama

SBS – Seychelles Bureau of Standards, Seychelles

WIS – Weizmann Institute of Science, Israel



**Figures**
**Figure 1** Map of locations of stations used in this study. The zoomed grid of the LMDz-
INCA model is also plotted with the NASA Shuttle Radar Topographic Mission (SRTM)
1km digital elevation data (DEM) as background (http://srtm.csi.cgiar.org). The grey shaded
area indicates the region with a horizontal resolution of $0.66° \times 0.51°$. The red close circle
(blue cross) represents the atmospheric station where flask (in-situ) measurements are
available and used in this study.

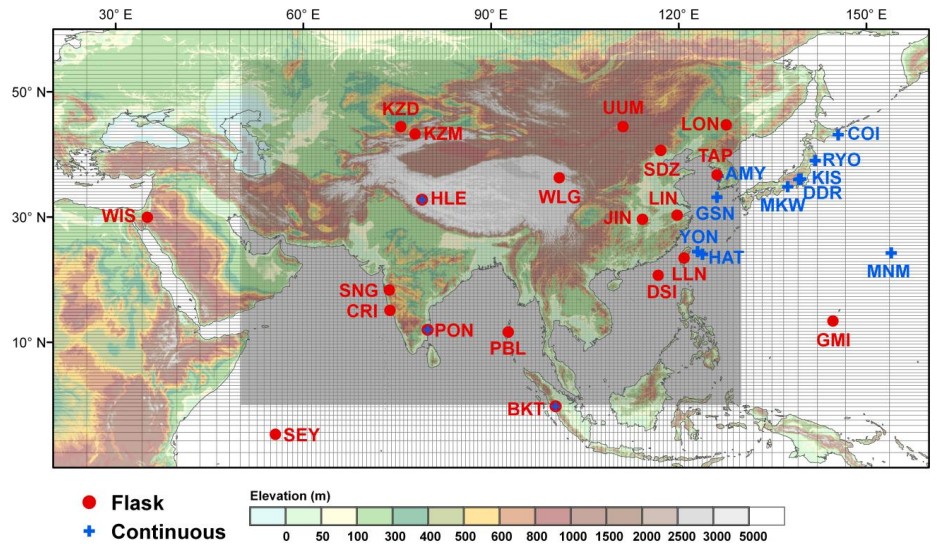






**Figure 2** Scatterplots of simulated and observed mean annual gradients of $CH_4$ (a, b) or $CO_2$
(c, d) between HLE and other stations. For both tracers, the simulated gradients are based on
simulations from ZASIA (a, c) and REG (b, d). In each panel, the black dotted line indicates
the identity line, whereas the grey solid line indicates the linear line fitted to the data. The
black (grey) texts give the mean bias (±1σ) and RMSE of the simulated mean annual
gradients in reference to the observed ones for stations within (outside) the zoomed region.
The italic type and open symbols in the legend denote stations outside the zoomed region.





**Figure 3** (a,c) Correlations between the observed and simulated $CH_4$ or $CO_2$ mean seasonal
cycles from ZASIA (y axis) and REG (x axis) for all available stations. (b,d) Ratios of the
simulated to observed $CH_4$ or $CO_2$ seasonal amplitude from ZASIA (y axis) and REG (x axis)
for all available stations. Stations within the dotted circles have a ratio of the simulated to
observed amplitude ranging 0.75–1.25 from both ZASIA and REG. The grey shaded regions
mark the domains where ZASIA better capture the seasonal amplitude than REG. For each
station, the mean seasonal cycle is calculated from the harmonics of the corresponding
smoothed fitting curve, and the seasonal amplitude is defined as the difference between the
seasonal maximum and minimum.



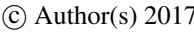



**Figure 4** Taylor diagrams showing correlations and normalized standard deviations (NSD;
the ratio of the simulated to observed standard deviation) between the simulated and observed
$CH_4$ (a,b) or $CO_2$ (c,d) synoptic variability for all available stations. For each station, the
synoptic variability is calculated from residuals from the smoothed fitting curve.




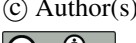



**Figure 5** Observed and simulated hourly $CH_4$ time series for AMY (126.32°E, 36.53°N,
133m a.s.l.) in Korean Peninsula and BKT (100.32°E, 0.20°S, 869m a.s.l.) in Indonesia. For
each panel, the black dots indicate the $CH_4$ measurements, while the red and blue dots
indicate the simulated $CH_4$ time series from ZASIA and REG, respectively. For BKT, we
also present the simulated $CH_4$ time series sampled at the first layer of both versions of the
model (colored in purple and green for ZASIA and REG, respectively).

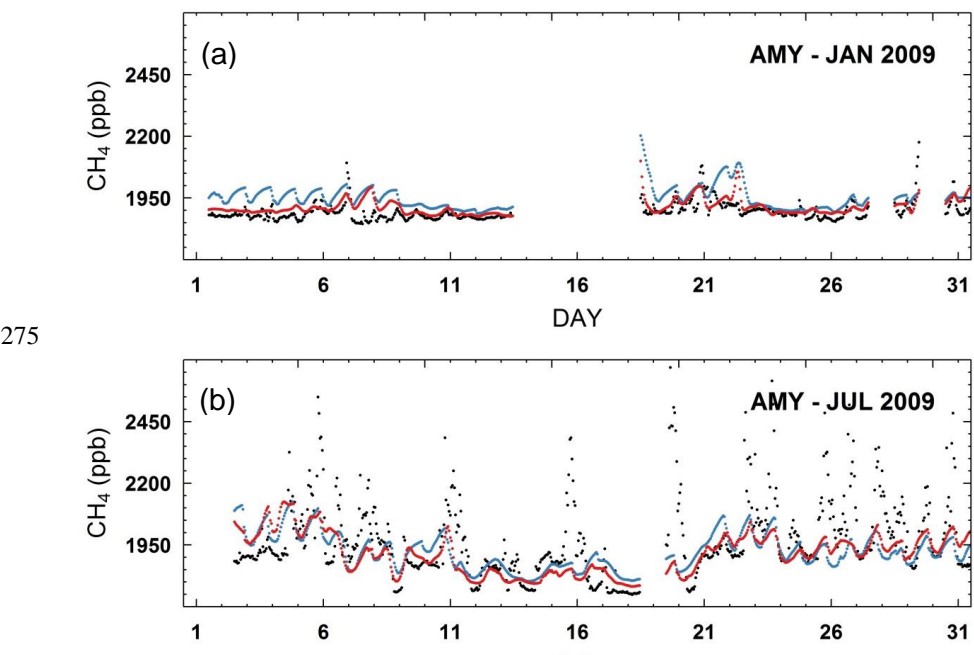







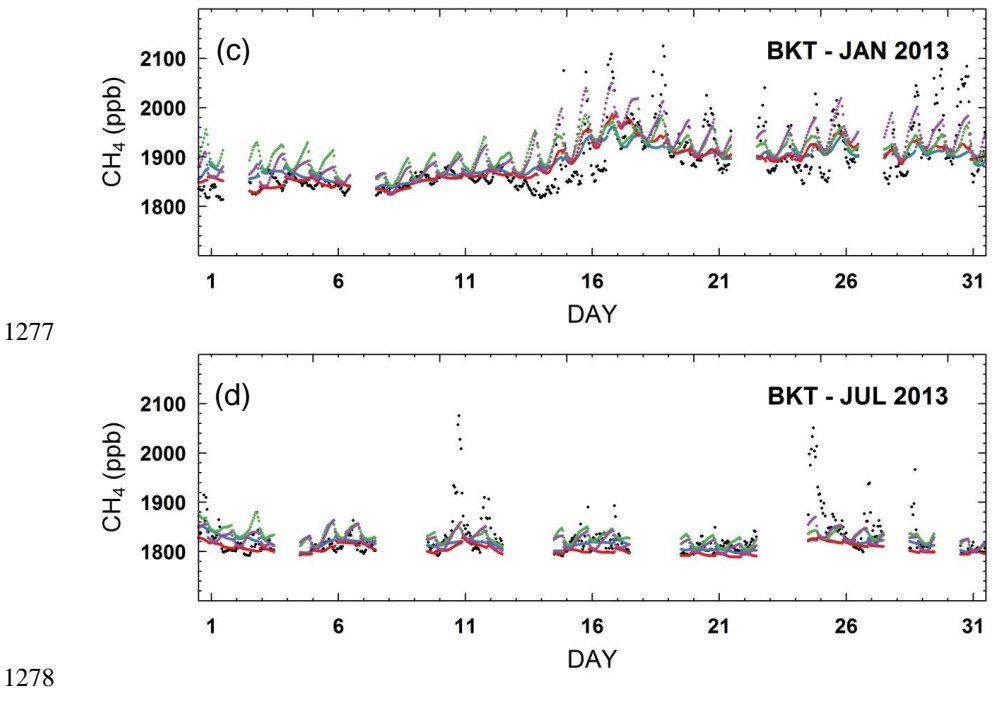






**Figure 6** Observed and simulated hourly $CO_2$ time series for GSN (126.12°E, 33.15°N, 144m
a.s.l.) in South Korea, PON (79.86°E, 12.01°N, 30m a.s.l.) in India and BKT (100.32°E,
0.20°S, 869m a.s.l.) in Indonesia. For each panel, the black dots indicate the $CO_2$
measurements, while the red and blue dots indicate the simulated $CO_2$ time series from
ZASIA and REG, respectively. For BKT, we also present the simulated $CO_2$ time series
sampled at the first layer of both models (colored in purple and green for ZASIA and REG,
respectively).

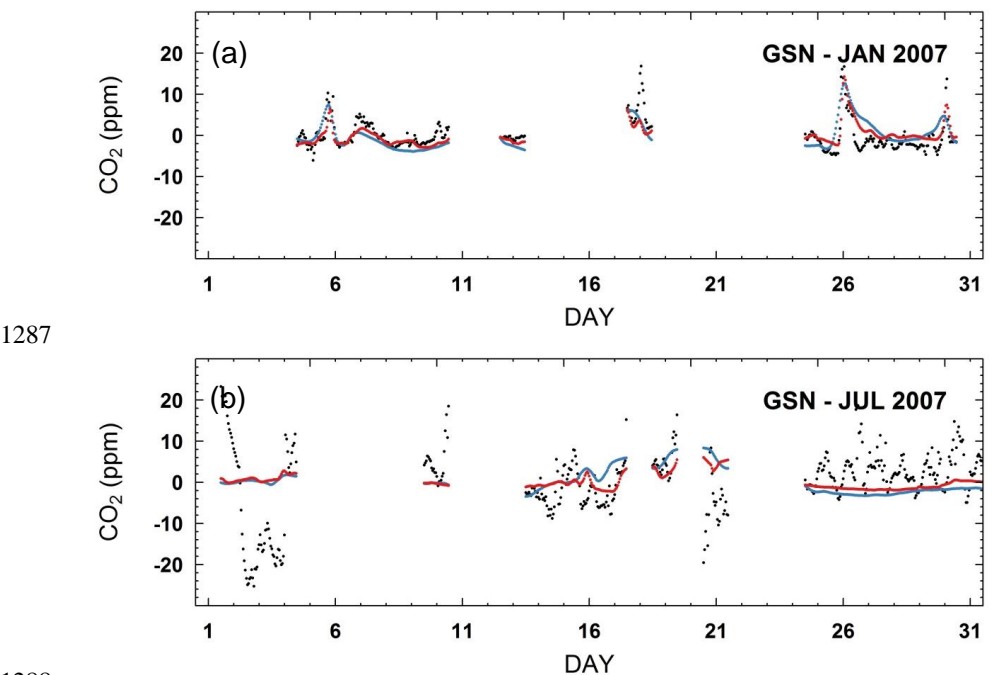















**Figure 7** Seasonal mean observed and simulated $CO_2$ vertical profiles over **(a)** East Asia
(EAS), **(b)** the Indian sub-continent (IND), **(c)** Northern Southeast Asia (NSA) and **(d)**
Southern Southeast Asia (SSA). The observed vertical profiles are based on $CO_2$ continuous
measurements onboard the commercial air flights from the CONTRAIL project during the
period 2006–2011. For each 1-km altitude bin and each subregion, the observed and
simulated time series are detrended (denoted as $\Delta CO_2$) and seasonally averaged during
January–March (JFM), April–June (AMJ), July–September (JAS) and October–December
(OND). For each panel, the error bars and shaded areas give the standard deviations of the
observed and simulated $\Delta CO_2$ at each altitude bin and within each subregion.

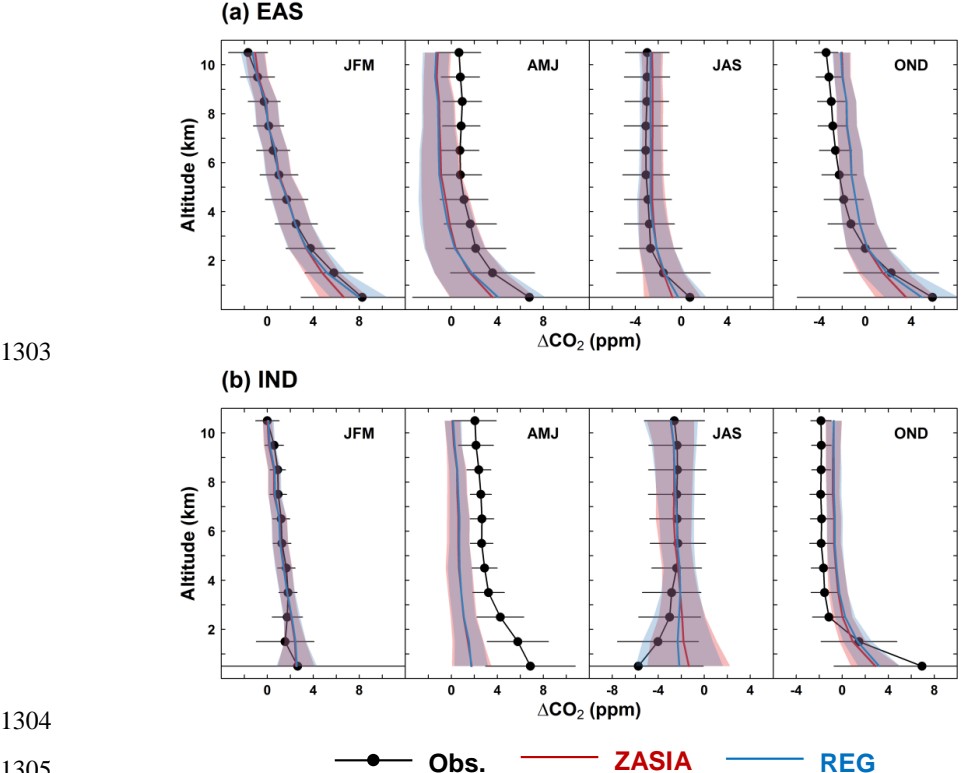








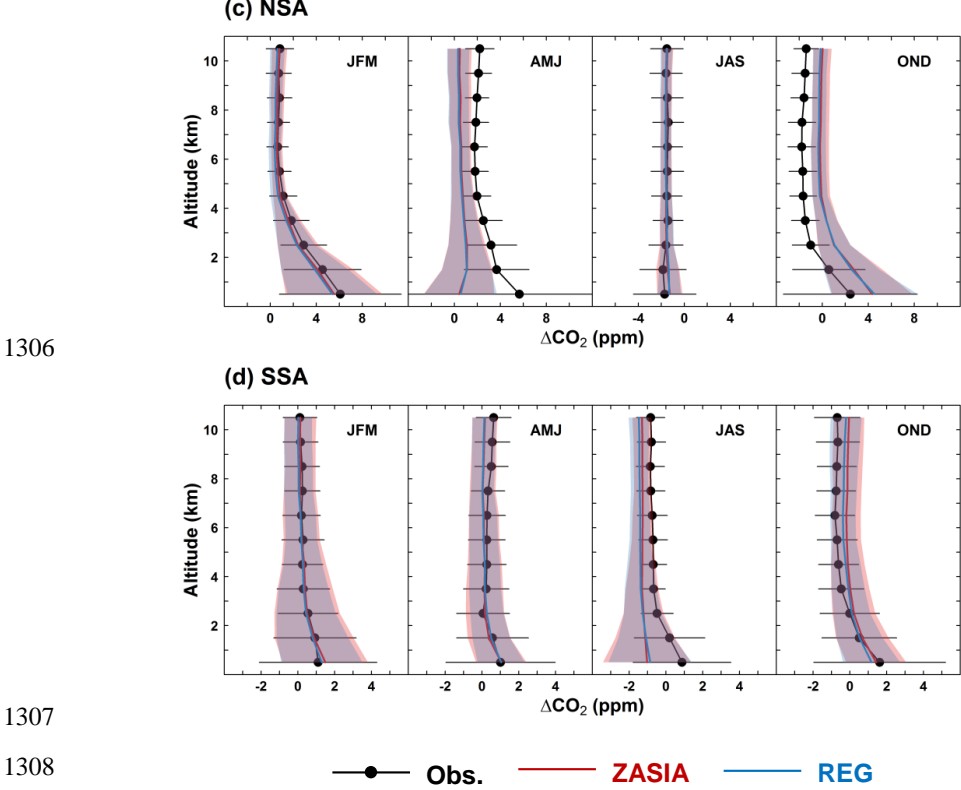








**Figure 8** Monthly mean observed and simulated $CO_2$ gradient between 1 and 4km over **(a)**
East Asia (EAS), **(b)** the Indian sub-continent (IND), **(c)** Northern Southeast Asia (NSA) and
**(d)** Southern Southeast Asia (SSA). For each subregion, the monthly $CO_2$ gradients are
calculated by averaging over all the vertical profiles the differences in $CO_2$ concentrations
between 1 and 4km. The error bars and shaded areas indicate the standard deviations of the
observed and simulated $CO_2$ gradients.

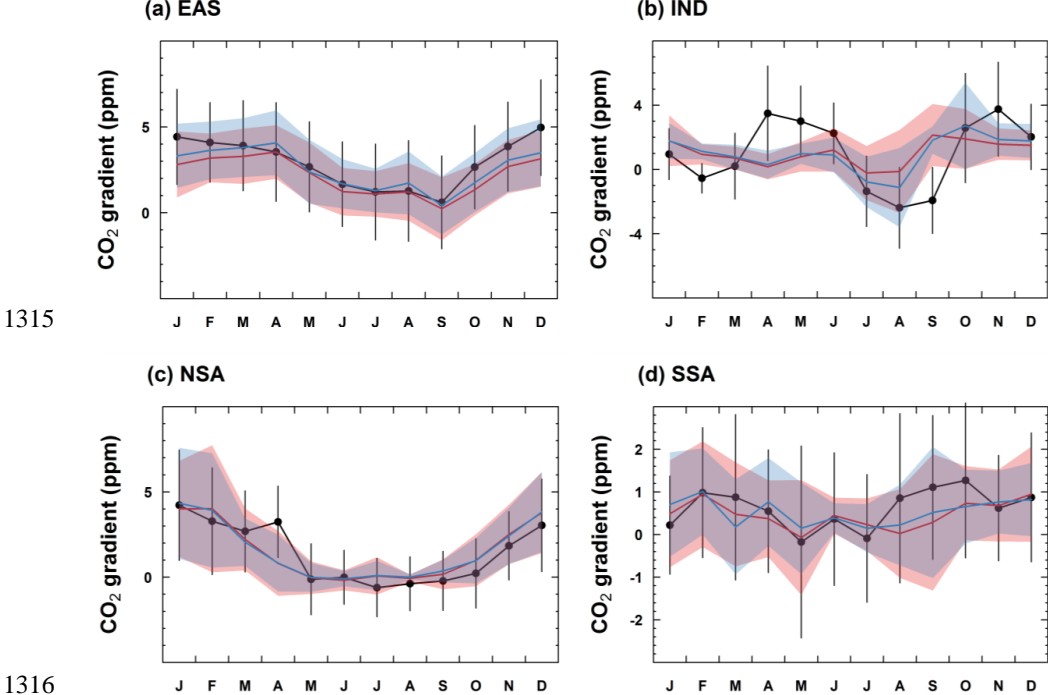

