# Peer review of "Simulating CH4 and CO2 over South and East Asia using the zoomed"

_Atmospheric Chemistry and Physics, 2016_

## Referee Comment (RC1) · Anonymous Referee #1 · 20 Apr 2017

This manuscript describes a detailed sensitivity simulation of CH4/CO2 in Asia with respect to horizontal resolution employing two different versions of the LMDzINCA model. This kind of study can be expected to contribute significantly to improving performance of data assimilation and accuracy of inverse modeling as the authors emphasize. The overall text is well written, and the authors very carefully discuss the results. However, most of the descriptions in this paper appear to be too detailed and sometime tedious although they may be needed to convey useful information to the data assimilation procedure. The subject of this paper seems to be appropriate to the ACP. However, I would like the authors to consider my questions and revise the manuscript before I recommend the publication of this paper. Details of my comments will be found in the

following.

Major Comments:

*M1: For "abstract" and "conclusions" section, I'm not convinced about conclusions of this manuscript. The authors state that the finer horizontal resolution version improves Asian $CH_4/CO_2$ simulation only moderately. Are you saying that enhancing horizontal resolution is not that useful (not beneficial)? I think you could more clearly express the message/implication of this study at least in abstract and conclusions parts.

*M2: This study just showed that a finer horizontal resolution more or less contributes to improvement of $CH_4/CO_2$ simulation for Asia. But it is very unclear whether this improvement is really significant or meaningful in terms of regional budget and flux estimate. I think the authors should check the impacts of other factors (at least vertical resolution or NEE) on the simulation as well as horizontal resolution for more clearly appealing the advantages of your zoomed method in the LMDzINCA modeling framework.

*M3: For the moderate improvement with ZASIA, I do not yet understand the reason for it. The authors give several potential candidates like matching between the model's grid and observation site, different transport, etc. But how much do they contribute? Or what is the most possible reason for the improvement?

*M4: The authors stated that the ZASIA version does not deteriorate the performance of $CH_4/CO_2$ outside the zoomed area (L383). But they seem to be looking only at the sites displayed in Figure 1 (mostly in Japan). How about the impacts on performance for other sites like in EU, US, Africa, and the southern hemisphere? This point should be clarified in the main text with an additional figure as supplementary material.

Minor Comments:

** L158 to L173: How do you represent diurnal variation in OH?

** L177 "The spin-up time of 6 years": Don't you have any trend or drift of global mean

CH4 concentration during this 6 years?

** L179 "already realistic": What do you mean by "realistic"? You should explain more about the initial conditions for CH4.

** L395 "better description of the surface fluxes and/or transport": Given the fact that CO2 simulation is not improved by ZASIA, the improvement seen in CH4 seems to be resulting from non-transport process (surface fluxes?).

** L435: There appears no explanation for the abbreviation of "NEE".

** L500 "rather coarse (19 layers)": How do you get the model concentrations at the elevation of the observational site? The model layers are linearly interpolated?

---

## Referee Comment (RC2) · Anonymous Referee #2 · 28 Apr 2017

This study presents a detailed comparison between CO2 and CH4 simulations from de LMDzINCA model and the available measurements over South East Asia. It is meant as a first step in preparation for flux inversions, to identify observed signals pointing to short-comings in the a priori fluxes or transport model uncertainty, in support of the inversion set-up. To this end a comparison is made between different model versions, and the added value of increased model resolution is assessed. The manuscript is well written, and the difference between model and measurements is carefully assessed. However, there should be a more efficient way to arrive it the main conclusions, e.g. by summarizing the performance in only a few key figures. This would also help to make the final conclusions more quantitative. In its current form, the scientific message is

not so clear. In my opinion, publication in ACP would require more than just model performance documentation. Therefore, additional effort is needed to strengthen the scientific significance of this work.

GENERAL COMMENTS

The conclusions describe the performance of the two model versions in qualitative, and sometimes rather vague, terms such as 'generally capable', 'moderately improves', 'fairly well', etc. Some key numbers are needed quantifying the performance, and the significance of performance differences. For example, one would expect improved resolution to pay out more on performance metrics addressing short-term variability, or at sides that are more influenced by small scale variability in the sources and sinks. Different temporal scales are addressed separately, but, to improve the scientific significance, the relation between them could be addressed in further detail.

The idea to compare $CO_2$ and $CH_4$ is interesting, however, it is difficult to compare the model performance between the two. It is like comparing apples and oranges, since the spatio-temporal scales that are influenced by the emissions of these tracers in relation to variations due to transport are so different. It is suggested that the emission uncertainty is more important for $CO_2$ than for $CH_4$, but because of the correlation between flux and transport uncertainties (e.g. the rectifier in case of $CO_2$) it is not possible to really separate these influences. If without this problem, the question remains what it means for the potential of the inversion to contribute to our understanding of the fluxes. The results suggest that this potential is better for $CO_2$, whereas I don't think this can really be objectively quantified just from forward simulations.

I see the value of assessing the benefits of improving model resolution. The trouble here, however, is the limiting vertical resolution. The conclusion that this resolution needs to be improved seems quite obvious to me, to the extent that I even wonder why this was not done from the start. It seems a necessary prerequisite for assessing the benefits of improved model resolution.
[Figure]

Why has the vertical profile comparison to CONTRAIL been limited to CO2. It is true that CH4 was measured only on a small subset of samples, but to include this could nevertheless be important to separate the influence of diurnal variations in emissions and PBL mixing. To me it seems that there is also some unexplored potential comparing diurnal cycle mismatches between CH4 (PBL mixing controlled) and CO2 (PBL mixing and flux variation controlled).

Since the aim was to prepare for inversions, what are the implications of this study for the inversion setup? I mean, the implications that are mention don't seem to have any practical consequences (except for the need for improved vertical resolution).

SPECIFIC COMMENTS

line 163: How is the OH scaling done? A single scaling factor?

line 194: Given the inter-annual variability of biomass burning, wouldn't it be better to use a climatological mean emission distribution for the extrapolated years?

section 2.2: Differences between calibration scales are mentioned but except for AMY CH4 it is not clear how these differences have been accounted for.

line 364: Is this after subtracting longer term components?

line 408: Given the short regional transport times it is unclear how errors in OH could play a role.

line 627: Would the improvements in PBL dynamics that are mentioned work in the right direction?

Table 1: How about the seasonal variation in anthropogenic CH4 emissions? (why are they taken into account for CO2 but not for CH4?). How about the temporal variability of biomass burning. It seems relevant to make use of available information regarding its sub-monthly variability, in particular when assessing the impact of improved resolution is an important goal.

**[ACPD](ACPD)**
TECHNICAL CORRECTIONS

Line 132: 'representthe'

Line 612: 'Here', where?

S4: Why does the legend show blue colors? It would be better to leave this part out given that positive hotspots are in blue also.

---

## Author Response (AR1)

This manuscript describes a detailed sensitivity simulation of $CH_4/CO_2$ in Asia with respect to horizontal resolution employing two different versions of the LMDzINCA model. This kind of study can be expected to contribute significantly to improving performance of data assimilation and accuracy of inverse modeling as the authors emphasize. The overall text is well written, and the authors very carefully discuss the results. However, most of the descriptions in this paper appear to be too detailed and sometime tedious although they may be needed to convey useful information to the data assimilation procedure. The subject of this paper seems to be appropriate to the ACP. However, I would like the authors to consider my questions and revise the manuscript before I recommend the publication of this paper. Details of my comments will be found in the following.

**[Response]** Thank you very much for your careful review and comments. Following the reviewers' suggestions, we launched new simulations with 39 vertical layers (L39) for both standard and zoom models, as compared to the previous simulations with only 19 vertical layers (L19). We updated the biomass burning emissions to the latest GFEDv4.1 for both $CH_4$ and $CO_2$ simulations. For $CH_4$, we also ran sensitivity test simulations, in which anthropogenic and wetland emissions are prescribed with the latest EDGARv4.3.2 and model outputs from ORCHIDEE. For $CO_2$, sensitivity test simulations are also performed with daily and 3-hourly biomass burning emissions from GFEDv4.1 (Table R1). We have rewritten most part of the results, discussions and conclusions accordingly. We also replied to your major and minor comments in the following, and hopefully our responses and revision adequately address all your comments and questions.

**Major Comments:**

M1: For "abstract" and "conclusions" section, I'm not convinced about conclusions of this manuscript. The authors state that the finer horizontal resolution version improves Asian $CH_4/CO_2$ simulation only moderately. Are you saying that enhancing horizontal resolution is not that useful (not beneficial)? I think you could more clearly express the message/implication of this study at least in abstract and conclusions parts.

**[Response]** Not really. The model's capability to represent the $CH_4$ or $CO_2$ variability at stations does not only depend on model resolution. In this paper we would like to more emphasize that, with finer model resolution, the model performance is more sensitive to accuracy of the prescribed surface fluxes, particularly distribution of sources/sinks at fine scales and their short-term variabilities. The sensitivity test simulations we launched for the revised paper also show importance of the flux data quality in model performance and thus benefits of improved model resolution. Following your suggestion, we revised the manuscript and clarify it in Abstract and Conclusion.

M2: This study just showed that a finer horizontal resolution more or less contributes to improvement of $CH_4/CO_2$ simulation for Asia. But it is very unclear whether this improvement is really significant or meaningful in terms of regional budget and flux estimate. I think the authors should check the impacts of other factors (at least vertical resolution or NEE) on the simulation as well as horizontal resolution for more clearly appealing the advantages of your zoomed method in the LMDzINCA modeling framework.

[Response] As we stated in Introduction, the number of regional ground stations in South and East Asia has increased during the recent decades. Observations from these stations will provide useful constraints on regional flux estimates, if gradients between stations and their variabilities can be well represented in transport models. Compared to the global transport model with rather coarse model resolution, the zoomed transport model used in our study has the potential to better capture the observed spatial and temporal variations at regional stations due to the reduced representation errors. The impact of model resolution on regional budget and flux estimate should be addressed by inverse modeling, which is beyond the scope of this study. Following your suggestion, we launched new simulations with 39 vertical layers (L39) for both standard and zoom models, as compared to the previous simulations with only 19 vertical layers (L19). For $CH_4$, we also ran sensitivity test simulations, in which anthropogenic and wetland emissions are prescribed with the latest EDGARv4.3.2 and model outputs from ORCHIDEE (Table R1). Detailed results and discussions are presented in Section 3 in the revised manuscript.

M3: For the moderate improvement with ZASIA, I do not yet understand the reason for it. The authors give several potential candidates like matching between the model's grid and observation site, different transport, etc. But how much do they contribute? Or what is the most possible reason for the improvement?

[Response] With the zoomed model, the explanation for the improved model performance on $CH_4$ mean annual gradients really depends on different stations. As mentioned in Section 3.1.1, the better performance at SDZ (117.12°E, 40.65°N, 293m a.s.l.) is more related to the detailed description of source distribution around the station; for the two coastal stations PON (79.86°E, 12.01°N, 30m a.s.l.) and CRI (73.83°E, 15.08°N, 66m a.s.l.), the improved model performance is related to the better characterization of the complex terrain (coastal topography) as well as the fluxes.

M4: The authors stated that the ZASIA version does not deteriorate the performance of $CH_4/CO_2$ outside the zoomed area (L383). But they seem to be looking only at the sites displayed in Figure 1 (mostly in Japan). How about the impacts on performance for other sites like in EU, US, Africa, and the southern hemisphere? This point should be clarified in the main text with an additional figure as supplementary material.

[Response] Following your suggestions, we further included several global/regional stations in Europe (the stations JFJ and MHD), North America (the stations ALT, BRW, NWR and

MLO), and the southern hemisphere (the stations AMS, CGO, and SPO) in this study (Table 2). Analyses show that the zoom versions do not deteriorate model performance outside the zoomed region compared to the standard versions. For example, the $CH_4$ and $CO_2$ annual gradients between HLE and these added stations can be well captured by both standard and zoom model versions (see open circles in Figure 2). Detailed results and discussions are presented in Section 3 and the supplementary material.

**Minor Comments:**

L158 to L173: How do you represent diurnal variation in OH?

[**Response**] As described in Section 2.1.1, we used climatological monthly OH concentration fields in this study and didn't consider the diurnal variation in OH fields. According to Patra et al. (2009), the $CH_4$ chemical lifetime in the troposphere is much longer than the dynamical residence time due to atmospheric transport, and accounting for OH diurnal cycle is not crucial for simulating seasonal, synoptic, and diurnal variations in $CH_4$ concentration fields.

L177 "The spin-up time of 6 years": Don't you have any trend or drift of global mean $CH_4$ concentration during these 6 years?

[**Response**] Take the global background station Mauna Loa as an example, Figure R1 presents time series of the simulated and observed $CH_4$ concentrations over the period 2000–2013, as well as the corresponding long term trends extracted from the data using the CCGVU curve fitting routine (Thoning et al., 1989). During the 6-year spin-up period (2000–2005), the simulated $CH_4$ concentrations decreased for the first three years and then levelled off. Drift of the global mean is found for both standard and zoom models, equivalent to around -12 ppb over this period. The model-observation disagreement in trend and global mean $CH_4$ concentrations results from the imperfect surface emissions and OH fields prescribed in the simulations. As we reply to the Reviewer #2 (Specific comments, Line 163), in this paper we are more focusing on the improvement gained from refinement of model grids rather than accurately reproducing the observed $CH_4$ concentrations and their interannual variations. Furthermore, all the traits and metrics we have considered to evaluate the model performance (i.e., annual mean gradient, seasonal cycle, synoptic variability, diurnal cycle and vertical gradient) give "relative" values that are not affected by the absolute $CH_4$ concentrations. Therefore the trend and drift of global mean $CH_4$ during the spin-up period will not have significant impact on comparison of performance between the standard and zoom models.

L179 "already realistic": What do you mean by "realistic"? You should explain more about the initial conditions for $CH_4$.

[**Response**] In the revised paper, the initial $CH_4$ concentration field we used for the updated simulations is defined based on the optimized initial state from a $CH_4$ inversion that assimilates observations from 50+ global background stations over the period 2006–2012

(Locatelli, 2014; Locatelli et al., 2015). The optimized initial $CH_4$ concentration field for the year 2006 was rescaled to the levels of the year 2000 and used as the initial state in our simulations. As the initial condition for $CH_4$ is optimized with observations, we assume it to be "realistic". Following your suggestion, we revised Section 2.1.1 accordingly to clarify the setup of initial condition for $CH_4$.

L395 "better description of the surface fluxes and/or transport": Given the fact that $CO_2$ simulation is not improved by ZASIA, the improvement seen in $CH_4$ seems to be resulting from non-transport process (surface fluxes?).

[**Response**] Here we mean that, with ZASIA, the model improvement on the $CH_4$ annual gradient at the stations SDZ, PON and CRI may "result from a reduction in representation error with a higher model horizontal resolution in the zoomed region, through a better description of the surface fluxes and/or transport around these stations". In fact, we also found improved model performance on the $CO_2$ annual gradients at the three stations, although not as significant as it is for $CH_4$ (Table R2). Therefore the model improvement may result from better characterization of either surface fluxes or transport processes or both.

L435: There appears no explanation for the abbreviation of "NEE".

[**Response**] Following your suggestions, we provide the full name (net ecosystem exchange) when the abbreviation is used for the first time.

L500 "rather coarse (19 layers)": How do you get the model concentrations at the elevation of the observational site? The model layers are linearly interpolated?

[**Response**] As described in Section 2.3, the modelled concentrations are sampled at the nearest gridpoint and vertical level to each station.

**Tables**

**Table R1** Model setups for different simulations.

| Simulation Code | Version | Anthrop. Emis. | Wetland Emis. |
|---|---|---|---|
| ST19_ED42 | 144×142 Standard, 19 layers | EDGAR4.2FT2010 | KAPLAN climatology |
| ZA19_ED42 | 144×142 Asian Zoom, 19 layers | | |
| ST39_ED42 | 144×142 Standard, 39 layers | | |
| ZA39_ED42 | 144×142 Asian Zoom, 39 layers | | |
| ST39_ED432 | 144×142 Standard, 19 layers | EDGAR4.3.2 | |
| ZA39_ED432 | 144×142 Asian Zoom, 19 layers | | |
| ST39_ED432ORC | 144×142 Standard, 39 layers | | ORCHIDEE climatology |
| ZA39_ED432ORC | 144×142 Asian Zoom, 39 layers | | |

**Table R2** The observed and simulated mean annual gradient of $CH_4$ **(a)** and $CO_2$ **(b)** between HLE and two stations (CRI, PON and SDZ) within the zoomed region. The bias reduction rates (in percentage) by using ZA compared to ST are also given for both 19- and 39-layer simulations.

a)

| $CH_4$ | OBS (ppb) | ST19 (ppb) | ZA19 (ppb) | Bias reduction | ST39 (ppb) | ZA39 (ppb) | Bias reduction |
|---|---|---|---|---|---|---|---|
| CRI | 17.5±12.7 | 9.3±4.1 | 20.2±7.1 | 66.6% | 8.6±3.0 | 23.0±6.7 | 38.8% |
| PON | 32.4±12.4 | 2.5±11.6 | 31.1±7.7 | 95.6% | 0.4±11.9 | 34.1±7.8 | 94.7% |
| SDZ | 90.0±15.4 | 125.1±18.8 | 86.8±16.0 | 91.0% | 128.5±19.3 | 100.4±22.4 | 73.0% |

b)

| $CO_2$ | OBS (ppm) | ST19 (ppm) | ZA19 (ppm) | Bias reduction | ST39 (ppm) | ZA39 (ppm) | Bias reduction |
|---|---|---|---|---|---|---|---|
| CRI | 4.6±0.9 | 1.2±0.1 | 2.0±0.3 | 25.5% | 1.4±0.1 | 2.2±0.2 | 25.2% |
| PON | 2.7±1.6 | 1.3±0.3 | 1.8±0.5 | 35.2% | 1.5±0.3 | 1.9±0.5 | 37.0% |
| SDZ | 6.8±0.5 | 8.8±1.3 | 7.7±1.9 | 57.9% | 9.3±1.5 | 8.1±2.3 | 48.1% |

**Figures**

**Figure R1** Time series of observed and simulated CH$_4$ concentrations at Mauna Loa (MLO, 19.54°N, 155.58°W, 3397) during the period 2000–2013. The simulated CH$_4$ concentrations are based on outputs from both standard (ST39ED42, blue circles) and zoom models (ZA39ED42, red circles). The solid lines indicate the corresponding long-term trends extracted from the data using the CCGVU curve-fitting routine (Thoning et al., 1989).

[Figure]

**References**

Locatelli, R.: Estimation des sources et puits de méthane: bilan planétaire et impacts de la modélisation du transport atmosphérique, Versailles-St Quentin en Yvelines, France. [online] Available from: http://www.theses.fr/2014VERS0035, 2014.

Locatelli, R., Bousquet, P., Saunois, M., Chevallier, F. and Cressot, C.: Sensitivity of the recent methane budget to LMDz sub-grid-scale physical parameterizations, Atmos. Chem. Phys, 15, 9765–9780, doi:10.5194/acp-15-9765-2015, 2015.

Patra, P., Takigawa, M., Ishijima, K., Choi, B.-C., Cunnold, D., J. Dlugokencky, E., Fraser, P., J. Gomez-Pelaez, A., Goo, T.-Y., Kim, J.-S., Krummel, P., Langenfelds, R., Meinhardt, F., Mukai, H., O'Doherty, S., G. Prinn, R., Simmonds, P., Steele, P., Tohjima, Y., Tsuboi, K., Uhse, K., Weiss, R., Worthy, D. and Nakazawa, T.: Growth rate, seasonal, synoptic, diurnal variations and budget of methane in the lower atmosphere, J. Meteorol. Soc. Japan, 87(4), 635–663, doi:10.2151/jmsj.87.635, 2009.

Thoning, K. W., Tans, P. P. and Komhyr, W. D.: Atmospheric carbon dioxide at Mauna Loa Observatory: 2. Analysis of the NOAA GMCC data, 1974–1985, J. Geophys. Res. Atmos., 94(D6), 8549–8565, doi:10.1029/JD094iD06p08549, 1989.

Anonymous Referee #2

This study presents a detailed comparison between $CO_2$ and $CH_4$ simulations from the LMDzINCA model and the available measurements over South East Asia. It is meant as a first step in preparation for flux inversions, to identify observed signals pointing to short-comings in the a priori fluxes or transport model uncertainty, in support of the inversion set-up. To this end a comparison is made between different model versions, and the added value of increased model resolution is assessed. The manuscript is well written, and the difference between model and measurements is carefully assessed. However, there should be a more efficient way to arrive it the main conclusions, e.g. by summarizing the performance in only a few key figures. This would also help to make the final conclusions more quantitative. In its current form, the scientific message is not so clear. In my opinion, publication in ACP would require more than just model performance documentation. Therefore, additional effort is needed to strengthen the scientific significance of this work.

**[Response]** Thank you very much for your careful review and comments. Following the reviewers' suggestions, we launched new simulations with 39 vertical layers (L39) for both STs and ZAs, as compared to the previous simulations with only 19 vertical layers (L19). We updated the biomass burning emissions to the latest GFEDv4.1 for both $CH_4$ and $CO_2$ simulations. For $CH_4$, we also ran sensitivity test simulations, in which anthropogenic and wetland emissions are prescribed with the latest EDGARv4.3.2 and model outputs from ORCHIDEE. For $CO_2$, sensitivity test simulations are also performed with daily and 3-hourly biomass burning emissions from GFEDv4.1 (Table R1). Following your suggestions, we have rewritten most part of results, discussions and conclusions in the manuscript accordingly. We also replied to your major and minor comments in the following, and hopefully our responses and revision adequately address all your comments and questions.

GENERAL COMMENTS

The conclusions describe the performance of the two model versions in qualitative, and sometimes rather vague, terms such as 'generally capable', 'moderately improves', 'fairly well', etc. Some key numbers are needed quantifying the performance, and the significance of performance differences. For example, one would expect improved resolution to pay out more on performance metrics addressing short-term variability, or at sites that are more influenced by small scale variability in the sources and sinks. Different temporal scales are addressed separately, but, to improve the scientific significance, the relation between them could be addressed in further detail.

**[Response]** Following your suggestion, we have rewritten the conclusions and implications. Key numbers are given with respect to the model improvement with finer horizontal resolution. We also claim that the performance of high resolution transport model is more sensitive to errors in meteorological forcings and surface fluxes, especially when short-term variabilities or stations close to source regions are examined. This emphasizes importance of accurate a priori $CH_4$ surface fluxes in high resolution transport modelling and inverse studies, particularly regarding locations and magnitudes of emission hotspots. Please refer to Section 4 for more details.

The idea to compare $CO_2$ and $CH_4$ is interesting, however, it is difficult to compare the model performance between the two. It is like comparing apples and oranges, since the spatio-temporal scales that are influenced by the emissions of these tracers in relation to variations due to transport are so different. It is suggested that the emission uncertainty is more important for $CO_2$ than for $CH_4$, but because of the correlation between flux and transport uncertainties (e.g. the rectifier in case of $CO_2$) it is not possible to really separate these influences. If without this problem, the question remains what it means for the potential of the inversion to contribute to our understanding of the fluxes. The results suggest that this potential is better for $CO_2$, whereas I don't think this can really be objectively quantified just from forward simulations.

**[Response]** We agree with Reviewer #2 that it's difficult to compare the model performance between $CO_2$ and $CH_4$, and that the correlation between flux and transport uncertainty is not possible to be really separated. In the revised paper we no longer suggest the emission uncertainty is more important for $CO_2$ than for $CH_4$. In fact, the emission uncertainty is important for both gases, yet in different ways. For $CH_4$, we highlight importance of uncertainty regarding the magnitudes and distribution of emission hotspots; while with respect to $CO_2$, we more focus on uncertainties related to the spatio-temporally varying NEE fluxes. We rephrased the conclusions and implications in Section 4 and removed statements about the potential of inversions to contribute to our understanding of $CO_2$ or $CH_4$ fluxes. However, in a few places we kept comparisons between $CO_2$ and $CH_4$ at specific stations. For example, in Section 3.2, the strong contrast in model performance between $CO_2$ and $CH_4$ seasonal cycles at BKT does suggest inaccurate seasonal variations in the prescribed $CO_2$ surface fluxes such as NEE.

I see the value of assessing the benefits of improving model resolution. The trouble here, however, is the limiting vertical resolution. The conclusion that this resolution needs to be improved seems quite obvious to me, to the extent that I even wonder why this was not done from the start. It seems a necessary prerequisite for assessing the benefits of improved model resolution.

**[Response]** Following your suggestion, we launched new simulations with 39 vertical layers (L39) for both STs and ZAs, as compared to the previous simulations with only 19 vertical layers (L19). The detailed model setups for control simulations and sensitivity tests prescribed with different surface fluxes are shown in Table R1. In brief, increasing model vertical resolution does not have as much impact on model performance as increasing the horizontal resolution at any temporal scale, although in several cases the combination of finer horizontal and vertical resolution tends to further increase the simulated amplitudes of variations (not necessarily improve the model performance). More detailed results and discussions are presented in Section 3.

Why has the vertical profile comparison to CONTRAIL been limited to $CO_2$? It is true that $CH_4$ was measured only on a small subset of samples, but to include this could nevertheless be important to separate the influence of diurnal variations in emissions and PBL mixing. To me it seems that there is also some unexplored potential comparing diurnal cycle mismatches between $CH_4$ (PBL mixing controlled) and $CO_2$ (PBL mixing and flux variation controlled).

**[Response]** We agree that the model-data comparison of vertical profiles for both $CO_2$ and $CH_4$ would be important to separate the influence of diurnal variations in surface fluxes and PBL mixing. The question here is that the vertical profiles from the CONTRAIL project are only limited to $CO_2$ measurements that are made by on-board continuous measurement equipment (CME). Measurements for $CH_4$ are also available, but they are only flask samples in the high troposphere and stratosphere. Please refer to Machida et al. (2008) for further information about the project and the dataset.

Since the aim was to prepare for inversions, what are the implications of this study for the inversion setup? I mean, the implications that are mention don't seem to have any practical consequences (except for the need for improved vertical resolution).

**[Response]** There are three implications for inversion setup, which we have elaborated in Section 4. First, the performance of high resolution transport model is more sensitive to accuracy of the prescribed surface fluxes, especially regarding locations and magnitudes of emission hotspots for $CH_4$. Therefore, one should be cautious when choosing an emission map as a priori for inversions. In particular, the unrealistic emission hotspots close to a station (as shown for UUM in Section 3.3.1) should be corrected, otherwise the inverted surface fluxes are likely to be strongly biased.

Second, as current bottom-up estimates of $CH_4$ sources and sinks still suffer from large uncertainties at fine scales, caution should be taken when one attempts to assimilate observations not realistically simulated by the high resolution transport model. These observations should be either removed from inversions or allocated with large uncertainties.

Third, representation of short-term variabilities is limited by model's ability to simulate boundary layer mixing and mesoscale transport in complex terrains. The recent implementation of new sub-grid physical parameterizations in LMDz is able to significantly improve simulation of the daily maximum during nighttime and thus diurnal cycles of tracer concentrations (Locatelli et al., 2015). To fully take advantage of high-frequency $CH_4$ or $CO_2$ observations at stations close to source regions, it is highly recommended to implement the new boundary layer physics in the current transport model, in addition to refinement of model horizontal and vertical resolutions. The current transport model with old planetary boundary physics is not capable to capture diurnal variations at continental or mountain stations, therefore only observations that are well represented should be selected and kept for inversions (e.g. afternoon measurements for continental stations and nighttime measurements for mountain stations).

SPECIFIC COMMENTS

Line 163: How is the OH scaling done? A single scaling factor?

[**Response**] In the revised paper, we relaunch $CH_4$ simulations with different model versions, using OH fields regridded from outputs of a full chemistry INCA with model grids of $96\times95\times39$. We don't scale the OH fields as did before to match the simulated global $CH_4$ growth rate with the observed one, as we are more focusing on the improvement gained from finer model resolutions rather than accurately reproducing the observed $CH_4$ concentrations and their interannual variations. Furthermore, all the traits and metrics we have considered to evaluate the model performance (i.e., annual mean gradient, seasonal cycle, synoptic variability, diurnal cycle and vertical gradient) give "relative" values that are not affected by the absolute $CH_4$ concentrations. Therefore the influences of the OH fields on the model improvement are assumed to be very small, and we don't scale them in the current $CH_4$ simulations. We revised the description of the OH fields accordingly in Section 2.1.1.

Line 194: Given the inter-annual variability of biomass burning, wouldn't it be better to use a climatological mean emission distribution for the extrapolated years?

[**Response**] In the updated simulations, we used GFEDv4.1 for emissions from biomass burning that are available over the whole running period (2000–2013). We revised the description of the prescribed surface fluxes accordingly in Section 2.1.2.

Section 2.2: Differences between calibration scales are mentioned but except for AMY $CH_4$ it is not clear how these differences have been accounted for.

[**Response**] As we described in Section 2.2, the $CH_4$ measurements at AMY are reported on the KRISS scale and they are not traceable to the WMO scale. For analyses of the $CH_4$ annual gradients between stations, we discard AMY because calibrations scales for different stations (i.e. AMY and HLE in this case) should be consistent for the calculation of gradients between them. For the analyses of seasonal cycle, synoptic variability and diurnal cycle, since the calibration scale doesn't significantly impact the results, we keep this station in these analyses.

Line 364: Is this after subtracting longer term components?

[**Response**] Yes. When we evaluated the model performance on $CH_4$ and $CO_2$ diurnal cycle, for each station daily means are subtracted from the raw data to remove any influence of interannual, seasonal or even synoptic variations. We revised Section 2.4.4 in the manuscript to clarify it.

Line 408: Given the short regional transport times it is unclear how errors in OH could play a role.

[**Response**] The main sink of $CH_4$ is oxidation by OH in the troposphere. Although we agree that the regional transport time is much shorter compared to the $CH_4$ lifetime, the spatial (both horizontally and vertically) and seasonal distribution of OH can influence the model performance on $CH_4$ annual gradients between stations and seasonal cycles. Here in the paper, the $CH_4$ annual gradient between TAP and HLE is significantly overestimated by both STs and ZAs. The overall poor performance at this station suggests the prescribed surface emissions are probably overestimated over the station's footprint area (also shown by overestimation of seasonal amplitude at TAP), yet errors in OH distribution may also play a role – although we are not clear about the magnitude. To address the question we need an inverse system that can optimized the OH fields by assimilating observations of a tracer with well-known fluxes (e.g., methylchloroform), which is beyond the scope of this study.

Line 627: Would the improvements in PBL dynamics that are mentioned work in the right direction?

[**Response**] Yes. In Locatelli et al. (2015) the authors evaluated the impact of new physical parameterizations recently implemented in LMDz on representation of trace gas transport and chemistry. These development and modification on physical parameterization are to improve simulation of vertical diffusion, mesoscale mixing by thermal plumes in the planetary boundary layer (PBL), and deep convection in the troposphere. Regarding the PBL dynamics, the thermal plume model is developed and combined with Yamada (1983) diffusion scheme to improve representation of the diurnal cycles of thermodynamical and dynamical variables of the boundary layer and of shallow cumulus clouds (Hourdin et al., 2002; Rio et al., 2008). Locatelli et al. (2015) showed that implementing this new PBL physics in LMDz significantly improves representation of the daily peak values of [222]Rn concentrations at continental stations compared to the old model version (see Figure 3 in their paper), and the simulated diurnal cycles can agree very well with the observed one at a few tested stations (e.g. Heidelberg, as shown in Figure 4 in their paper). So far we haven't implemented the new PBL physics in our current model simulations, we will explore its potential in representation diurnal cycle of $CO_2$ and $CH_4$ in future studies.

Table 1: How about the seasonal variation in anthropogenic $CH_4$ emissions? (why are they taken into account for $CO_2$ but not for $CH_4$?). How about the temporal variability of biomass burning? It seems relevant to make use of available information regarding its sub-monthly variability, in particular when assessing the impact of improved resolution is an important goal.

[**Response**] For the first question, we have considered the seasonal variation in anthropogenic $CH_4$ emissions from rice cultivation based on Matthews et al. (1991), as described in Section 2.1.2 and Table 1. The seasonal variations for other emission sectors are much smaller compared to those from rice paddies, and monthly sector-specific dataset is currently not available for the whole study period. Therefore we didn't considered seasonal variations in $CH_4$ emissions from those sectors. We revised Section 2.1.2 to further clarify it.

For the second question, in this study we used monthly biomass burning dataset from the GFEDv4.1 product. We agree that including its sub-monthly variability would be relevant when assessing the impact of increased resolution on model performance, especially for those stations that are potentially influenced by episodic large biomass burning events. Following your suggestion, we launched sensitivity test simulations for $CO_2$ using daily and 3-hourly biomass burning emissions for the year 2013, and evaluate the model performance on synoptic variation and diurnal cycle at a tropical station located in western Indonesia BKT (100.32°E, 0.20°S, 869m a.s.l.). Results show that simulations prescribed with daily or 3-hourly variability of biomass burning do not always improve representation of $CO_2$ diurnal cycle at BKT – sometimes could be worse, which again emphasizes uncertainties in prescribed surface fluxes (including uncertainties in temporal variability) as one of major factors that influence the model performance.

TECHNICAL CORRECTIONS

Line 132: 'representthe'

[**Response**] We corrected it.

Line 612: 'Here', where?

[**Response**] We rewrote the whole section. Please refer to Section 3.4.1.

S4: Why does the legend show blue colors? It would be better to leave this part out given that positive hotspots are in blue also.

[**Response**] Following your suggestion, we corrected the legend in Figure S4.

**Tables**

**Table R1** Model setups for different simulations.

[revised manuscript text omitted]